# Tuning the Anthranilamide Peptidomimetic Design to Selectively Target Planktonic Bacteria and Biofilm

**DOI:** 10.3390/antibiotics12030585

**Published:** 2023-03-15

**Authors:** Rajesh Kuppusamy, Muhammad Yasir, Tsz Tin Yu, Florida Voli, Orazio Vittorio, Michael J. Miller, Peter Lewis, David StC Black, Mark Willcox, Naresh Kumar

**Affiliations:** 1School of Chemistry, The University of New South Wales (UNSW), Sydney, NSW 2052, Australia; 2School of Optometry and Vision Science, The University of New South Wales (UNSW), Sydney, NSW 2052, Australia; 3Children’s Cancer Institute, Lowy Cancer Research Centre, University of New South Wales, Sydney, NSW 2052, Australia; 4School of Clinical Medicine, University of New South Wales, Sydney, NSW 2052, Australia; 5ARC Centre of Excellence in Convergent Bio-Nano Science and Technology, University of New South Wales, Sydney, NSW 2052, Australia; 6School of Environmental and Life Sciences, College of Engineering, Science and Environment, The University of Newcastle, Newcastle, NSW 2308, Australia; 7Hunter Biological Solutions Pty Ltd., Newcastle, NSW 2310, Australia

**Keywords:** peptidomimetics, biofilm, anthranilamide

## Abstract

There is a pressing need to develop new antimicrobials to help combat the increase in antibiotic resistance that is occurring worldwide. In the current research, short amphiphilic antibacterial and antibiofilm agents were produced by tuning the hydrophobic and cationic groups of anthranilamide peptidomimetics. The attachment of a lysine cationic group at the tail position increased activity against *E. coli* by >16-fold (from >125 μM to 15.6 μM) and greatly reduced cytotoxicity against mammalian cells (from ≤20 μM to ≥150 μM). These compounds showed significant disruption of preformed biofilms of *S. aureus* at micromolar concentrations.

## 1. Introduction

There is an urgent need to combat the emergence of multi-drug-resistant (MDR) bacteria [1,2]. One of the most widespread drug-resistant bacterial strains is methicillin-resistant *Staphylococcus aureus* (MRSA), which poses a major threat to human health, particularly in hospitals but also in the wider community [3]. In the last several decades, newly approved antibiotics have mostly been structural derivatives of existing drugs, which appear to offer a small window of efficacy before there is a significant increase in the frequency of bacterial resistance to them [4].

Resistance can be mediated by mutations in the genes of microbial systems, the antibiotic targets, or the acquisition of new genes from other microbes. Another mechanism that microbes use to resist the action of antimicrobials is the development of sessile communities called biofilms [5,6,7]. The host immune system is unable to combat biofilm-associated infections due to the presence of a thick layer of exopolysaccharides (EPS) [8]. Horizontal gene transfer is higher in biofilms than in planktonic cells [9], which can increase the speed of the spread of resistance. Out of five major anti-biofilm mechanisms reported in the literature, the most prevalent mechanism is the disruption or degradation of the membrane potential of biofilm-embedded cells [10]. Biofilms of Gram-positive *S. aureus* and Gram-negative *Escherichia coli* cause several infections in humans [11].

In recent years, antimicrobial peptides (AMPs) have attracted significant attention as a new generation of antibiotics due to their low propensity to induce resistance [12,13]. AMPs are rich in both hydrophobic and cationic residues, and their positive charge allows them to attack bacterial membranes which are more negatively-charged relative to eukaryotic membranes [14,15]. However, AMPs are susceptible to proteolytic breakdown causing unexpected pharmacokinetics and toxicity, which has prevented their entry into the clinic [16]. This has stimulated the development of AMP-mimicking molecules, or “peptidomimetics”, which retain the balance and spatial arrangement of the hydrophobic and hydrophilic groups of the natural peptides. Examples of peptidomimetics that have been studied in the literature include α-peptides [17], β-peptides [18,19], peptoids [20], biphenyl-based peptidomimetics [21], and biaryl 1,2,3-triazolyl peptidomimetics [22]. Some of these peptidomimetics showed significant anti-biofilm activity against Gram-positive and Gram-negative bacteria [23,24,25,26,27,28,29,30].

There is still an urgent need to design and develop peptidomimetics that can combat biofilm-related infections but show minimal eukaryotic cytotoxicity. Our previous work has shown that anthranilamide peptidomimetic derivatives can function as effective antibacterial and biofilm disruptors [23]. The peptidomimetic compounds bearing primary amine cationic groups showed better antibacterial activity compared to those with tertiary ammonium or quaternary ammonium groups. However, the amine compounds were cytotoxic against mammalian cells.

The current research describes the development of biofilm-disrupting and non-toxic anthranilamide peptidomimetics by exploring the structure–activity relationship of the scaffold shown in Figure 1. In series I, the hydrophobic group attached to the anthranilamide core was varied between naphthoyl and various related cyclic and heterocyclic compounds. The naphthoyl group is frequently found as a hydrophobic group in peptidomimetic compounds showing good activity against Gram-positive and Gram-negative bacteria [31,32,33,34]. In series II, the bromo substituent was replaced with different phenyl-substituted groups to generate biaryl derivatives, which are common bioactive motifs [22,32,35,36]. The amino acid lysine is a frequently used cationic group in peptidomimetics with antimicrobial [37,38,39,40,41] and antibiofilm activity [2]. Hence, the series III compounds contained lysine cationic groups attached to the tail of the anthranilamide peptidomimetic compounds. The hydrophobic and hydrophilic properties of tryptophan make it ideal for insertion into bacterial cell membranes [21,42]. In all the series of compounds, the tryptophan was kept constant.

## 2. Results and Discussion

### 2.1. Synthesis of Peptidomimetic Compounds (Series I–III)

Series I compounds: The key intermediates **6a** and **6b** for new peptidomimetic derivatives were synthesized based on our previous publication as described in (Figure 1). Carboxybenzyl (Cbz)-protected tryptophan **1** was reacted with monoprotected diamines **2a**,**2b** under 1-ethyl-3-(3-dimethylaminopropyl) carbodiimide (EDCI) coupling conditions to give amides **3a**–**3b**. These were subjected to hydrogenation using 10% Pd/C to remove the Cbz-protecting group, giving peptides **4a**–**4b** with a free N-terminal amine group. The **4a**–**4b** were utilized for the ring-opening reaction with 5-bromoisatoic anhydride (**5**) to yield the intermediate amine **6a**–**6b**.

The acid chlorides **7a**–**7k** were generated from the corresponding commercially available carboxylic acids (**7e′** and **7f′** prepared using hydroxy derivative; detailed procedure in Appendix A) as shown in Figure 2 and were directly used in situ in the subsequent step. The acid chlorides **7a**–**7k** were used to install the hydrophobic group on the benzene ring of **6a** and **6b** (**10a**–**10k** from **6a**, **11a**–**11e** from **6b**). Finally, the deprotection of the terminal boc-protected amino group followed by trituration with diethyl ether and few drops of methanol gave the desired series I compounds **12a**–**12k**, **13a**–**13e** as shown in Figure 3.

Series II compounds: The intermediate **6a** was reacted with different boronic acids **14a**–**14d** using Pd(PPh_3_)_4_ catalyst to give the biaryl products **15a**–**15d** (Figure 4). These were deprotected using trifluoroacetic acid (TFA) to yield the corresponding amines **16a**–**16d**. The parent analogue **17** was synthesized by deprotecting 6a directly without performing the Suzuki–Miyaura cross-coupling reaction.

#### Series III Compounds

The series I compounds **12a**, **12j**–**12k** were treated with Boc-protected hydroxy succinimide ester (Boc-Lys(Boc)-OSu) in the presence of triethylamine to give **18a**–**18c** (Figure 5 which then deprotected using TFA to give the series III compounds **19a**–**19c.**

### 2.2. Antibacterial Activity of Peptidomimetic Compounds (Series I–III)

The antibacterial activity screening was performed for the series I–III compounds (Table 1). In series I, most of the compounds showed moderate to good antibacterial activity against *S. aureus* (MIC = 3.9–15.6 µM), except **12f**–**12h** (125 μM) bearing indole or thiophene as hydrophobic groups. This suggested that heterocyclic hydrophobic groups may not be ideal for antibacterial activity. For the naphthoyl-based hydrophobic groups, those that were attached at the 2-position, such as **12a** (3.9 μM [23]) and **12d** (7.8 μM), usually showed better antibacterial activity compared to their corresponding 1-substituted counterparts **12b** (15.6 μM) and **12c** (15.6 µM). Compounds **12i**–**12k** bearing the biphenyl hydrophobic group, which is a bio isostere of the naphthoyl group, showed good antibacterial activity. Among these, the 3-substituted biphenyl compound **12j** (3.9 µM) showed better antibacterial activity than the 2- and 4-biphenyl substituted **12i** (15.6 µM) and **12k** (7.8 µM) compounds.

However, compounds **12a**–**12k** did not show antibacterial activity against *E. coli* even at concentrations >125 µM. Interestingly, compounds **13c** and **13d** bearing methoxy substituents on their naphthoyl groups showed moderate antibacterial activity against *E. coli* (both 62.5 µM).

The series II compounds **16a**–**16d** showed good to moderate antibacterial activity against *S. aureus*. Compounds **16c** and **16d** bearing electron-withdrawing substituents showed significant antibacterial activity. Moreover, replacing with tert-butyl substituted phenyl ring **16a** (7.8 µM) and a bulky naphthyl group substituted compound **16b** (3.9 µM) improved the antibacterial activity by four-fold against *S. aureus* and also showed moderate activity against *E. coli*. Taken together, these results indicated the importance of having a bulky hydrophobic group for improving staphylococcal activity. Comparing the series I and series II compounds, the potency of the compounds against both Gram-positive and Gram-negative increased with increasing bulkiness and net positive charge. The series I compound **12a** showed MIC_90_ of 3.9 µM but only against *S. aureus*, however the series II compound 16b by replacing the same hydrophobic group to increase the net positive charge showed antibacterial activity against *S. aureus* (3.9 µM) and *E. coli* (31.2 µM), respectively. Overall, the series II compounds had greater antibacterial activity than the series I compounds.

The series III compounds **19a**–**19c** were tested for antibacterial activity against *S. aureus* and *E. coli* (Table 2). All the compounds showed moderate antibacterial activity against both bacterial strains compared to **12a**, **12j**, and **12k**.

Interestingly, the lysine derivatives **19a** and **19c** with a naphthyl hydrophobic group and meta-substituted biphenyl group showed >16-fold improvement in antibacterial activity against *E. coli* (15.6 µM) compared to the parent compound **12a**, **12j** (>125 µM), although its activity against *S. aureus* (15.6 µM) was four-fold worse. Taken together, these results suggest that the number of cationic charges as well as the positioning of the hydrophobic group can affect antibacterial activity against both *S. aureus* and *E. coli*. All the antibacterial activity of compounds was compared with colistin. These peptidomimetic compounds showed good antibacterial activity against Gram-positive (*S. aureus*) compared to colistin. However, colistin is good against Gram-negative (*E. coli*) bacteria.

### 2.3. Cytoplasmic Membrane Permeability Studies

Compounds **13d** (series I), **16b** (series II), **19a** (series III) and **19b** (series III) were selected as representative examples of each series with strong antibacterial activities (particularly against *S. aureus*). As AMPs are known to be membrane-targeting, membrane permeability assays are frequently used to confirm the mode of action of peptidomimetics [43]. In this assay, *S. aureus* and *E. coli* were treated with the active compounds in the presence of the membrane potential-sensitive cyanine dye 3,3**′**-dipropylthiadicarbocyanine iodide (DiSC3(5)). Perturbation of the membrane by the compounds leads to loss of the membrane potential gradient (depolarization), causing the dye to be released into the medium and resulting in an increase in fluorescence intensity.

The membrane disruption of active compounds (**13d**, **16b**, **19a**, **19b**) at 2× MIC against *S. aureus* and *E. coli* is shown in Figure 2. **13d** was relatively inactive in this test compared to other compounds. Compound **19a** showed greater membrane disruption in *E. coli* than in *S. aureus*. In contrast, compound **19b** showed a reversed trend. Though the cationic group is similar the hydrophobic group is different (Naphthyl vs. Biphenyl) and it may play a role in less membrane permeability. All compounds depolarized the membrane after 3 min incubation, with most showing maximum activity after 6 min, after which there was a plateau in activity.

### 2.4. Membrane Integrity Studies

The mechanism of action of compounds **16b**, **19a,** and **19c** was further studied using *Bacillus subtilis* strain BS23, which contains a fusion of green fluorescence protein (GFP) to the α-subunit of the membrane-localized ATP synthase [44,45]. In the assay, if the compounds damage the membrane, the green fluorescence will change from a uniform distribution to a clustered distribution. Epifluorescence microscopy was used to image the changes in the cells. As shown in Figure 3, compounds **16b**, **19a**, and **19b** (50 μM) all caused membrane damage as seen by clustering of the dye, similar to the known membrane-targeting antibiotic colistin.

### 2.5. Biofilm Disruption Studies

Bacteria in biofilms are 10–1000 times more resistant to conventional antibiotics than planktonic bacteria. Moreover, biofilms are insensitive to antiseptics and many host immune responses. Since up to 80% of all microbial infections are biofilm-related [46], an effective antibacterial agent should also be able to disrupt established biofilms to tackle bacterial infections. Hence, the active compounds (**13d**, **16b**, **19a**, and **19b**) were tested for their ability to disrupt established *S. aureus* and *E. coli* biofilms using the crystal violet staining assay.

The most active compound against *S. aureus* was the biaryl derivative **16b**, which disrupted 93% of biofilm at 4× MIC (15.6 µM) concentration (Figure 4). The series I compound **13d** disrupted biofilms by 75% at 4× MIC (15.6 µM) and 99% at 8× MIC (31.2 µM). The series III compounds **19a** and **19b** disrupted biofilms by over 60% at 2× MIC (31.2 µM), while at 8× MIC (125 µM) they eradicated almost 99% of the biofilm. Overall, compounds **16c**, **19a,** and **19b** showed the greatest ability to disrupt biofilms, which could be due to their 2+ net charge compared to the 1+ charge of **13f**.

The compounds showed a reduced ability to disrupt *E. coli* biofilms compared to *S. aureus* biofilms. Only compounds **13d** and **19b** showed significant disruption of the *E. coli* biofilms (Figure 5). Interestingly, compound **13d** disrupted 59% of *E. coli* biofilm mass at 2× MIC (62.4 µM) concentration whereas compound **19b** showed little effect, which was the reverse order of potency compared to *S. aureus* biofilms. Compounds **16b** and **19a** were not effective against *E. coli* biofilm. In planktonic bacteria, most AMPs act through the disruption of cytoplasmic membranes [47]. However, against biofilms, there is little evidence that membrane disruption is the mechanism of action [48]. Though compounds **19a** and **19b** contained two positive net charges, they did not disrupt *E. coli* biofilm as effectively as compound 13d having simple amine with methoxy substituent. Hence, a net charge might not be the only consideration for biofilm disruption ability.

### 2.6. Cytotoxicity Assay

In cytotoxicity studies, all the compounds in series III showed less cytotoxicity against MRC5 human fibroblasts compared to the series I and series II compounds. Compound **19b** showed a very wide therapeutic window as its IC_50_ to human cells (164 μM) was over 10 times the MIC_90_ (15.6 μM) for either *S. aureus* or *E. coli*. Compound **19c** showed similarly excellent selectivity for *S. aureus*, but only moderate activity against *E. coli.*

## 3. Structure–Activity Relationship Studies

Based on the biological results, structure-activity relationships (SARs) can be deduced for these three series of anthranilamide peptidomimetic compounds. For the series I compounds, a change in the nature of the hydrophobic group as well as its position of attachment had a profound effect on their antibacterial activities. Specifically, compounds with the naphthoyl substituent attached at the 2-position showed better antibacterial activity than those attached at the 1-position. Moreover, heterocyclic hydrophobic groups were generally not preferred. The compounds with one-carbon linker compounds showed similar or two- to three-fold increase in antibacterial activity compared to the two-carbon linker compounds. The series I compounds were much more effective against *S. aureus* (Gram-positive bacteria) than *E. coli* (Gram-negative bacteria).

In the series II compounds, the hydrophobic group was moved away from the aniline functionality and inserted in the place of the bromine on the anthranilamide core. These compounds, which were also dicationic, showed very good antibacterial activity against *S. aureus* along with low cytotoxicity against mammalian cells. Specifically, the bulkiness of the hydrophobic group played an important role towards the antibacterial activity of these compounds compared to their electronic effects. The compound with naphthoyl hydrophobic group showed very good antibacterial activity against *S. aureus* and *E. coli*.

Finally, the series III compounds investigated the effect of retaining the hydrophobic group at the aniline site but increasing the cationic charge by attaching a lysine moiety to the peptide tail. Interestingly, this resulted in compounds with the greatest antibacterial activity against *E. coli* and with low cytotoxicity.

## 4. Materials and Methods

### 4.1. Biological Assays

#### 4.1.1. Minimum Inhibitory Concentration (MIC)

The antimicrobial activity of the compounds was evaluated by a broth microdilution assay using the procedure described by CLSI. Briefly, bacteria were grown to mid-log phase in Muller Hinton broth (MHB) with shaking at 120 rpm and incubated at 37 °C for 12–16 h. Following incubation, bacteria were washed three times in PBS pH 7.4 at 3500 g for 10 min. After washing, bacteria were diluted with fresh MHB. The turbidity of the bacterial suspensions was adjusted so that OD_660_nm was 0.1, which gave 1 × 10^8^ CFU/mL, and then further diluted to achieve 5 × 10^5^ CFU/mL as a final bacterial concentration. Each compound was diluted (250–3.9 µM) through two-fold dilution. Wells in microtiter plates were loaded with 100 µL of inoculum containing 5 × 10^5^ CFU/mL bacteria. Wells without any compound and containing only bacteria were used as negative controls (i.e., no inhibition of growth). Wells with media only were set as blank. The microtiter plate was wrapped with paraffin to prevent evaporation and incubated with shaking at 120 rpm and 37 °C for 18–24 h. After incubation, spectrophotometric readings were taken. The well without any bacterial growth and showing zero spectrophotometric reading was regarded as the MIC of the compounds.

#### 4.1.2. Cytoplasmic Membrane Permeability Assay

The method was adopted from Wu et al. [49] with slight modifications. Bacterial cytoplasmic membrane permeability was determined using membrane potential sensitive dye diSC3-5 (3,3′-dipropylthiadicarbocyanine iodide) which penetrates inside bacterial cells depending on the membrane potential gradient of the cytoplasmic membrane. Bacteria were grown in MHB to mid-log phase by incubating with shaking at 37 °C for 18–24 h. Following incubation, bacteria were washed with 5 mM HEPES containing 20 mM glucose pH 7.2 and resuspended in the same buffer to an OD_600_ 0.05–0.06 which gave 1 × 10^7^ CFU/mL. The dye diSC3-5 was added at 4 µM to the bacterial suspension. The suspensions were incubated at room temperature for 1 h in the dark for maximum dye take-up by the bacterial cells. Then, 100 mM KCl was added to balance the K+ outside and inside the bacterial cell to prevent further uptake or outflow of the dye. For Gram-negative bacteria, 0.5 mM EDTA was used to destabilize the lipopolysaccharides-Mg^2+^-Ca^2+^ complex to help in dye penetration without affecting bacterial growth. A total of 100 µL of bacterial suspension was added to 96-well microtiter plate and with equal volume of antimicrobial compounds. DMSO (20%) was set as a positive control while dye and only bacterial cells were set as a negative control. Fluorescence was measured with a luminescence spectrophotometer at 3 min intervals at an excitation wavelength of 622 nm and an emission wavelength of 670 nm.

#### 4.1.3. Membrane Integrity Studies

Bacillus subtilis BS23 (atpA-GFP) was grown in LB (10 g/L Tryptone, 5 g/L Yeast Extract, 5 g/L NaCl) supplemented with 0.5% (*w*/*v*) xylose at 37 °C, to an OD_600_ of 0.3 and then treated with 50 μM of the AMP mimics, 10 μg/mL of colistin or 0.25% (*v*/*v*) DMSO (vehicle control). The cells were grown for a further 45 min before being examined by epifluorescence microscopy. Micrographs were background subtracted, merged, and aligned in images.

#### 4.1.4. Biofilm Disruption Assay

Bacterial cultures (*S. aureus* and *E. coli*) were grown in MHB media overnight at 37 °C with shaking at 120 rpm. Cultures were diluted (1:20) in an MHB medium and 200 μL aliquots were dispensed to flat bottom 96-well plate wells (Sarstedt, Mawson Lakes, Australia). Cultures were supplemented with varying concentrations of synthetic compounds dissolved in DMSO. Biofilm was grown in a 96-well plate for 24 h followed by the addition of synthetic compounds and incubated further for 24 h. Plates were sealed with self-adhesive microplate sealers (TopSeal-A, PerkinElmer) to allow air diffusion and to prevent condensation. Biofilms adhered on polystyrene substratum were quantified by crystal violet staining as described previously. The experiment was performed in triplicate.

#### 4.1.5. Toxicity Assay

Normal human lung fibroblasts MRC-5 were cultured in minimal essential medium (MEM, Invitrogen) supplemented with 10% foetal calf serum (FCS), 1% L-glutamine–penicillin–streptomycin, 2% sodium bicarbonate, 1% non-essential amino acids (NEAA), and 1% sodium pyruvate. The cell line was maintained at 37 °C in 5% CO_2_ as an adherent monolayer and was passaged upon reaching confluence by standard cell culture techniques. MRC-5 cells were seeded at 2 × 10^4^ cells per well in 96-well plates to ensure full confluence (quiescence). Cells were treated for 24 h after seeding with 0.1 to 1000 μM of compounds. After 72 h of drug incubation, the treated media was replaced with fresh media containing 10% Alamar Blue and the cells were incubated for another 6 h. The metabolic activity was detected by spectrophotometric analysis by assessing the absorbance of Alamar blue as previously described by Pasquier et al. Cell proliferation was determined and expressed as a percentage of untreated control cells. The determination of IC_50_ values was performed using GraphPad Prism 6 (San Diego, CA, USA). Each experiment was performed in triplicate and was repeated in three independent experiments.

### 4.2. General Notes—Synthesis

All chemical reagents were purchased from commercial sources (Combi-Blocks (San Diego, CA, USA), Chem-Impex (Wood Dale, IL, USA), and Sigma Aldrich (St. Louis, MO, USA)) and used without further purification. The solvents were commercial and used as obtained. The reactions were performed using oven-dried glassware under an atmosphere of nitrogen and in anhydrous conditions (as required). Room temperature refers to the ambient temperature. Yields refer to chromatographically and spectroscopically pure compounds unless otherwise stated. The reactions were monitored by thin-layer chromatography (TLC) plates that were pre-coated with Merck silica gel 60 F254. Visualization was accomplished with UV light, and a ninhydrin staining solution in n-butanol. Flash chromatography and silica pipette plugs were performed under positive air pressure using Silica Gel 60 of 230–400 mesh (40–63 μm) and also using Grace Davison LC60A 6-μm for reverse phase chromatography. Infrared spectra were recorded using a Cary 630 ATR spectrophotometer. High-resolution mass spectrometry was performed by the Bioanalytical Mass Spectrometry facility, UNSW. Proton and Carbon NMR spectra were recorded in the solvents that were specified using a Bruker DPX 300 or a Bruker Avance 400 or 600 MHz spectrometer as designated. Chemical shifts (δ) are quoted in parts per million (ppm), to the nearest 0.01 ppm and internally referenced relative to the solvent nuclei. ^1^HNMR spectroscopic data are reported as follows (chemical shift in ppm; multiplicity in br, broad; s, singlet; d, doublet; t, triplet; q, quartet; quint, quintet; sext, sextet; sept, septet; m, multiplet; or as a combination of these (e.g., dd, dt, etc.)); coupling constant (J) in hertz, integration, proton count, and assignment.

#### 4.2.1. Procedure A for Synthesis of **3a** and **3b**

To the stirred solution of an acid 1 (1.0 mmol), amine **2a** or **2b** (1.0 mmol), HOBt (1.0 mmol), DIEA (2.5 mmol) in DMF (5–10 mL) and EDCI (1.2 mmol) was added portion-wise. The reaction was stirred for 16 h and then water was added. The solid settled was filtered out and dried under vacuo to yield the desired products **3a** or **3b** as an off-white solid in good yields.

#### 4.2.2. Procedure B for Synthesis of **4a** and **4b**

To the stirred solution of **3a** or **3b** in THF, 10% Pd/C was added while purging nitrogen. The hydrogen balloon was fitted and degassed with hydrogen. The reaction mixture was stirred under a hydrogen atmosphere for 12 h and then filtered through celite bed and dried the solvent under reduced pressure to yield **4a** and **4b**. To the residue, dichloromethane was added and the solid settled as off-white solid and was filtered out and dried under vacuo.

#### 4.2.3. Procedure C for Synthesis of **6a** and **6b**

The suspension of isatoic anhydride (1 mmol) and compound **4a** or **4b** (1 mmol) in anhydrous acetonitrile (20 mL) was refluxed under an argon atmosphere for 16 h. After completion of the reaction, the mixture was concentrated in vacuo to yield the crude compound, which was subjected to trituration using acetonitrile and diethyl ether. The solid was filtered out and dried under vacuum to afford **11a** and **11b** as off-white solids.

#### 4.2.4. General Procedure F for Synthesis of **7a**–**7o**

Acid chlorides were generated using acid **7a’**-**7o’** (1.0 mmol) and oxalyl chloride (1.0 mmol) in dichloromethane (3.0 mL) with a catalytic amount of DMF (drop) for 1 h and concentrated under reduced pressure and taken immediately for next step.

#### 4.2.5. General Procedure G for Synthesis of **10a**–**10k** and **11a**–**11e**

To the stirred solution of amine **6a** or **6b** (0.36 mmol) and Et_3_N (1.08 mmol) in CH_2_Cl_2_ the in situ generated acid chlorides **7a**–**7k** in CH_2_Cl_2_ were added and stirred at rt for 12 h. The reaction mixture was diluted with ethyl acetate and washed with saturated NaHCO_3_, brine solution and dried under anhydrous Na_2_SO_4_ and concentrated under reduced pressure. The residue was triturated with acetonitrile and diethyl ether to yield the compounds **10a**–**10k**. The compounds **11a**–**11e** were prepared using the above procedure with amine **6b** and acid chlorides **7a**–**7e**.

#### 4.2.6. General Boc Deprotection Procedure H for Synthesis of **12a**–**12k** and **13a**–**13e**

To a solution of **10a**–**10k** or **11a**–**11e** (0.1 mmol) in dichloromethane (1.0 mL) was added TFA (1.0 mL) at 0 °C. The reaction mixture was warmed to room temperature and stirred for 6 h. After completion of the reaction, the solvent was removed under a reduced pressure and treated with diethyl ether and the solid filtered out and dried under high vacuum to yield the desired products.

#### 4.2.7. General Suzuki–Miyaura Cross-Coupling Procedure I for Synthesis of **15a**–**15d**

To the degassed solution of the bromo compound **6a** (0.5 mmol) and appropriate boronic acid (**14a**–**14f**) (0.6 mmol) in toluene and ethanol (5.0:5.0 mL), 2N Na_2_CO_3_ (1.5 mmol) and Pd (PPh_3_)_4_ (2.5 mol%) were added. The reaction mixture was heated at 80 °C for 12 h. The reaction mixture was filtered through celite and washed with ethyl acetate. The organic layer was diluted with water. The organic layer was separated and washed with brine solution and dried under anhydrous Na_2_SO_4_ and concentrated under reduced pressure. The residue was purified using silica gel column chromatography using hexane:ethyl acetate (50:50) as eluent.

#### 4.2.8. General Procedure for Synthesis of **16a**–**16d**

Following the general procedure H, the compounds **16a**–**16d** were synthesized from **15a**–**15d**.

#### 4.2.9. General Procedure J for Synthesis of **18a**–**18c**

To the solution of Boc-Lys(Boc)-Osu (1.0 mmol) and Et3N (4.0 mmol) in THF (5.0 mL) appropriate compound (1.0 mmol) (**12i**–**12k**) was added and the reaction mixture was stirred at room temperature for 12 h. The reaction mixture was diluted with ethyl acetate and water. The organic layer was separated and washed with brine solution and dried under anhydrous Na_2_SO_4_ and concentrated under reduced pressure. The residue was purified using silica gel column chromatography using hexane: ethyl acetate (50:50) as eluent to yield the products **18a**–**18c**.

#### 4.2.10. General Procedure for Synthesis of **20a**–**20d**

Following the general procedure H, the compounds **20a**–**20d** were synthesized from **19a**–1**9d**.

Analytical data:

The analytical data for intermediate up to **6a**, **6b**, and final compounds **10a**, **12a** were already mentioned in our previous publication [20].

methyl 2-methoxy-1-naphthoate (**6**).

The title compound **6** was prepared from compound **5** (3.0 g, 14.8 mmol) according to the general procedure C. The product **6** was obtained as an off-white solid (2.56 g, 80%); 1H NMR (400 MHz, Chloroform-d) δ 7.94–7.87 (m, 1H), 7.76 (ddt, J = 0.9, 8.6, 25.1 Hz, 2H), 7.50 (ddd, J = 1.4, 6.8, 8.4 Hz, 1H), 7.37 (ddd, J = 1.2, 6.8, 8.1 Hz, 1H), 7.29 (d, J = 9.1 Hz, 1H), 4.04 (s, 3H), 3.97 (s, 3H);13C NMR (100 MHz, Chloroform-d) 168.7, 154.6, 131.8, 131.1, 128.7, 128.6, 128.2, 127.8, 124.3, 123.9, 113.2, 56.9, 52.6; HRMS (ESI): *m*/*z* calcd for C13H12O3 [M + Na]+: 239.0679; found: 239.0679.

2-methoxy-1-naphthoic acid (**7c’**).

The title compound **7c’** was prepared from compound 6 (2.0 g, 9.25 mmol) according to the general procedure C. The product **7c’** was obtained as an off-white solid (1.6 g, 90%); 1H NMR (400 MHz, DMSO-*d*_6_) δ 13.19 (s, 1H), 8.02 (d, J = 9.0 Hz, 1H), 7.95–7.88 (m, 1H), 7.69 (dd, J = 1.0, 8.5 Hz, 1H), 7.58–7.47 (m, 2H), 7.40 (ddd, J = 1.2, 6.7, 8.1 Hz, 1H), 3.93 (s, 3H); 13C NMR (100 MHz, DMSO-*d*_6_) 168.5, 153.1, 130.8, 129.9, 128.1, 128.0, 127.5, 124.0, 123.4, 118.7, 113.8, 56.5; HRMS (ESI): *m*/*z* calcd for C12H10O3 [M + Na]+: 225.0522; found: 225.0521.

methyl 3-methoxy-2-naphthoate (**9**).

The title compound **9** was prepared from compound **8** (3.0 g, 14.8 mmol) according to the general procedure C. The product **9** was obtained as an off-white solid (2.7 g, 85%); 1H NMR (400 MHz, Chloroform-d) δ 8.36 (s, 1H), 7.87 (ddd, J = 0.6, 1.3, 8.2 Hz, 1H), 7.82–7.71 (m, 1H), 7.57 (ddd, J = 1.3, 6.9, 8.3 Hz, 1H), 7.43 (ddd, J = 1.2, 6.9, 8.2 Hz, 1H), 7.26 (s, 1H), 4.06 (s, 3H), 4.01 (s, 3H).;13C NMR δ C (101 MHz, Chloroform-d) 166.8, 155.9, 136.2, 132.9, 128.8, 128.6, 127.7, 126.6, 124.5, 121.8, 106.9, 56.1, 52.4; HRMS (ESI): *m*/*z* calcd for C_13_H_12_O_3_ [M + Na]+: 1008.4593; found: 239.0679.

3-methoxy-2-naphthoic acid (**7d’**).

The title compound **7d’** was prepared from compound **9** (2.0 g, 9.25 mmol) according to the general procedure C. The product **7d’** was obtained as an off-white solid (1.56 g, 89%); 1H NMR (400 MHz, DMSO-*d*_6_) δ 12.81 (s, 1H), 8.16 (s, 1H), 7.91–7.84 (m, 1H), 7.79 (dd, J = 0.9, 8.3 Hz, 1H), 7.49 (ddd, J = 1.4, 6.9, 8.2 Hz, 1H), 7.40–7.29 (m, 2H), 3.86 (s, 3H);13C NMR (100 MHz, DMSO-*d*_6_) 167.4, 154.8, 135.3, 130.9, 128.4, 128.0, 127.1, 126.5, 124.3, 123.6, 106.8, 55.7; HRMS (ESI): *m*/*z* calcd for C_12_H_10_O_3_ [M + Na]+: 225.0522; found: 225.0522.

tert-butyl (S)-(2-(2-(2-amino-5-bromobenzamido)-3-(1H-indol-3-yl)propanamido)ethyl)carbamate (**6a**).

The title compound **6a** was prepared from compound **5** (4.0 g, 8.65 mmol) and **4a** (2.8 g, 8.65 mmol) according to the general procedure E. The product **6a** was obtained as a grey solid (3.0 g, 65%); 1H NMR (400 MHz, DMSO-*d*_6_) δ 10.78 (s, 1H), 8.35 (d, J = 8.0 Hz, 1H), 8.10 (t, J = 5.8 Hz, 1H), 7.69 (dd, J = 5.1, 15.9 Hz, 2H), 7.31 (d, J = 8.1 Hz, 1H), 7.23 (dd, J = 2.4, 8.8 Hz, 1H), 7.17 (d, J = 2.5 Hz, 1H), 7.05 (t, J = 7.5 Hz, 1H), 6.98 (t, J = 7.4 Hz, 1H), 6.76 (t, J = 5.6 Hz, 1H), 6.62 (d, J = 8.9 Hz, 1H), 6.48 (s, 2H), 4.64–4.55 (m, 1H), 3.21–2.94 (m, 6H), 1.37 (s, 9H); ^13^C NMR(75 MHz, DMSO-*d*_6_) 172.4, 167.9, 156.1, 149.3, 136.5, 134.6, 131.1, 127.7, 123.9, 121.3, 118.9, 118.7, 118.6, 116.1, 111.7, 111.1, 105.2, 78.1, 54.5, 28.7, 27.8; HRMS (ESI): *m*/*z* calcd for C_25_H_30_BrN_5_O_4_ [M + Na]+: 566.1373; found: 566.1369.

tert-butyl (S)-(3-(2-(2-amino-5-bromobenzamido)-3-(1H-indol-3-yl)propanamido)propyl)carbamate (**6b**).

The title compound **6b** was prepared from compound **5** (4.0 g, 8.65 mmol) and **4b** (3.1 g, 8.65 mmol) according to the general procedure E. The product **6b** was obtained as a grey solid (2.8 g, 58%); ^1^H NMR (400 MHz, DMSO-*d*_6_) δ 10.78 (s, 1H), 8.36 (s, 1H), 8.02 (s, 1H), 7.73–7.64 (m, 2H), 7.30 (d, J = 7.7 Hz, 1H), 7.23 (d, J = 9.1 Hz, 1H), 7.17 (s, 1H), 7.05 (s, 1H), 6.98 (s, 1H), 6.75 (s, 1H), 6.62 (d, J = 10.2 Hz, 1H), 6.47 (s, 2H), 4.63–4.54 (m, 1H), 3.23–3.14 (m, 2H), 3.10–3.06 (m, 2H), 2.93–2.88 (m, 2H), 1.51–1.46 (m, 2H), 1.37 (s, 9H).;^13^C (100 MHz, DMSO-*d*_6_) 172.1, 167.9, 155.0, 149.3, 136.5, 131.0, 127.7, 123.9, 121.3, 118.9, 118.7, 118.6, 116.1, 111.7, 111.1, 105.3, 77.9, 54.6, 37.8, 36.7, 30.0, 28.7, 27.9; HRMS (ESI): *m*/*z* calcd for C_26_H_32_BrN_5_O_4_ [M + H]+: 558.1710; found: 558.1711.

The synthesis and analytical data for *tert*-butyl (S)-(2-(2-(2-(2-naphthamido)-5-bromobenzamido)-3-(1*H*-indol-3-yl)propanamido)ethyl)carbamate (**10a**) were published already [20].

tert-butyl (S)-(2-(2-(5-bromo-2-(2-(naphthalen-1-yl)acetamido)benzamido)-3-(1H-indol-3-yl)propanamido)ethyl)carbamate (**10b**).

The title compound **10b** was prepared from compound **6a** (0.3 g, 0.55 mmol) and **7b** (0.66 mmol; **7b** in situ preparation from **7b’** using procedure F) according to the general procedure G. The product **10b** was obtained as an off-white solid (270 mg, 70%); 1H NMR (400 MHz, DMSO-*d*_6_) δ 10.95 (s, 1H), 10.82 (s, 1H), 8.88 (d, J = 8.0 Hz, 1H), 8.24–8.15 (m, 2H), 8.04–7.96 (m, 1H), 7.94–7.86 (m, 1H), 7.86–7.78 (m, 2H), 7.69 (d, J = 7.6 Hz, 1H), 7.60 (dd, J = 2.4, 8.9 Hz, 1H), 7.54–7.30 (m, 5H), 7.15 (d, J = 2.3 Hz, 1H), 7.07 (ddd, J = 1.4, 7.0, 8.1 Hz, 1H), 7.00 (ddd, J = 1.2, 7.0, 8.0 Hz, 1H), 6.79 (t, J = 5.6 Hz, 1H), 4.70–4.55 (m, 1H), 4.16–4.05 (m, 2H), 3.25 (dd, J = 4.5, 14.7 Hz, 1H), 3.16–2.96 (m, 5H), 1.35 (s, 9H).;^13^C NMR (100 MHz, DMSO-*d*_6_) 171.7, 169.7, 167.0, 156.1, 138.0, 136.5, 134.6, 133.8 132.3, 131.7, 131.2, 128. 9, 128.7, 127.7, 126.7, 126.2, 126.0, 124.4, 124.1, 124.0, 122.7, 121.4, 118.9, 118.7, 114.8, 111.38, 110.9, 78.1, 60.2, 54.9, 42.2, 28.6, 27.8, 21.2; HRMS (ESI): *m*/*z* calcd for C_37_H_38_BrN_5_O_5_ [M + Na]+: 734.1949; found: 734.1950.

tert-butyl (S)-(2-(2-(5-bromo-2-(2-methoxy-1-naphthamido)benzamido)-3-(1H-indol-3-yl)propanamido)ethyl)carbamate (**10c**).

The title compound **10c** was prepared from compound **6a** (0.3 g, 0.55 mmol) and **7c** (0.66 mmol; **7c** in situ preparation from 7c’ using procedure F) according to the general procedure G. The product 1**0c** was obtained as an off-white solid (288 mg, 72%); ^1^H NMR (400 MHz, DMSO-*d*_6_) δ 11.54 (s, 1H), 10.79 (s, 1H), 9.01 (d, J = 7.9 Hz, 1H), 8.61 (d, J = 9.0 Hz, 1H), 8.11 (t, J = 5.6 Hz, 1H), 8.05 (d, J = 8.9 Hz, 1H), 8.00 (d, J = 2.4 Hz, 1H), 7.95–7.88 (m, 1H), 7.79 (ddd, J = 1.8, 8.7, 12.5 Hz, 2H), 7.64 (d, J = 7.8 Hz, 1H), 7.54–7.43 (m, 2H), 7.39 (ddd, J = 1.3, 6.8, 8.1 Hz, 1H), 7.29 (dt, J = 0.9, 8.1 Hz, 1H), 7.16 (s, 1H), 7.03 (ddd, J = 1.3, 7.0, 8.2 Hz, 1H), 6.95 (ddd, J = 1.1, 7.0, 7.9 Hz, 1H), 6.71 (t, J = 5.7 Hz, 1H), 4.59–4.50 (m, 1H), 3.78 (s, 3H), 3.26–3.17 (m, 1H), 3.15–2.85 (m, 5H), 1.33 (s, 9H), 1.23 (s, 2H);13C NMR (101 MHz, DMSO-*d*_6_) 171.3, 167.2, 165.4, 156.0, 153.8, 138.6, 136.5, 134.7, 131.9, 131.5, 131.0, 128.66, 128.6, 128.0, 127.6, 124.5, 124.0, 123.0, 122.8, 121.3, 120.5, 120.0, 118.8, 118.7, 115.0, 114.2, 111.7, 110.8, 77.9, 56.8, 54.9, 39.4, 38.9, 29.8, 27.7; HRMS (ESI): *m*/*z* calcd for C_37_H_38_BrN_5_O_6_ [M + Na]+: 750.1898; found: 750.1904.

tert-butyl (S)-(2-(2-(5-bromo-2-(3-methoxy-2-naphthamido)benzamido)-3-(1H-indol-3-yl)propanamido)ethyl)carbamate (**10d**).

The title compound **10f** was prepared from compound **6a** (0.3 g, 0.55 mmol) and **7f** (0.66 mmol; **7f** in situ preparation from **7f’** using procedure F) according to the general procedure G. The product **10f** was obtained as an off-white solid (260 mg, 65%); 1H NMR (400 MHz, DMSO-*d*_6_) δ 11.86 (s, 1H), 10.78 (s, 1H), 8.97 (d, J = 8.1 Hz, 1H), 8.65 (d, J = 9.0 Hz, 1H), 8.57 (s, 1H), 8.22 (t, J = 5.6 Hz, 1H), 8.03–7.97 (m, 1H), 7.88 (dd, J = 1.0, 8.4 Hz, 1H), 7.83 (d, J = 2.4 Hz, 1H), 7.75–7.63 (m, 2H), 7.58 (ddd, J = 1.3, 6.8, 8.3 Hz, 1H), 7.47 (s, 1H), 7.45–7.38 (m, 1H), 7.27–7.13 (m, 2H), 7.06–6.91 (m, 2H), 6.77 (t, J = 6.0 Hz, 1H), 4.75–4.62 (m, 1H), 3.88 (s, 3H), 3.27 (dd, J = 4.2, 14.9 Hz, 1H), 3.03–2.96 (m, 2H), 3.22–3.03 (m, 3H), 1.34 (s, 9H);^13^C NMR (100 MHz, DMSO-*d*_6_) 171.8, 167.0, 163.5, 156.1, 154.5, 137.9, 136.5, 136.1, 134.5, 133.3, 131.5, 129.3, 128.9, 127.9, 127.7, 127.7, 126.8, 125.2, 124.9, 124.0, 123.7, 123.5, 121.3, 118.8, 118.6, 114.8, 111.7, 110.9, 107.3, 78.1, 55.9, 55.0, 28.6, 27.9; HRMS (ESI): *m*/*z* calcd for C_37_H_38_BrN_5_O_6_ [M + Na]+: 750.1896; found: 750.1901.

tert-butyl (S)-(2-(2-(5-bromo-2-(quinoline-2-carboxamido)benzamido)-3-(1H-indol-3-yl)propanamido)ethyl)carbamate (**10e**).

The title compound **10e** was prepared from compound **6a** (0.3 g, 0.55 mmol) and **7e** (0.66 mmol; **7e** in situ preparation from **7e**’ using procedure F) according to the general procedure G. The product **10e** was obtained as an off-white solid (294 mg, 77%); 1H NMR (400 MHz, DMSO-*d*_6_) δ 12.96 (s, 1H), 10.76 (s, 1H), 9.01 (d, J = 8.0 Hz, 1H), 8.70 (d, J = 8.9 Hz, 1H), 8.63 (d, J = 8.5 Hz, 1H), 8.25 (t, J = 7.9 Hz, 2H), 8.15–8.05 (m, 2H), 8.00 (d, J = 2.5 Hz, 1H), 7.89 (ddd, J = 1.5, 6.8, 8.5 Hz, 1H), 7.82–7.69 (m, 3H), 7.27–7.20 (m, 2H), 7.03–6.89 (m, 2H), 6.74 (t, J = 5.9 Hz, 1H), 4.79 (dt, J = 7.1, 13.0 Hz, 1H), 3.30–3.25 (m, 1H), 3.24–3.09 (m, 3H), 3.04–2.94 (m, 2H), 1.32 (s, 9H);13C NMR δ C (101 MHz, DMSO-*d*_6_) 171.2, 166.6, 162.7, 155.6, 149.4, 145.8, 138.3, 137.5, 136.0, 134.5, 131.3, 130.7, 129.4, 129.0, 128.5, 128.1, 127.3, 123.6, 123.5, 121.9, 120.9, 118.6, 118.4, 118.2, 114.6, 111.3, 110.5, 77.6, 54.6, 28.2, 27.4; HRMS (ESI): *m*/*z* calcd for C_35_H_35_BrN_6_O_5_ [M + Na]+: 721.1745; found: 721.1747.

tert-butyl (S)-(2-(2-(5-bromo-2-(1H-indole-2-carboxamido)benzamido)-3-(1H-indol-3-yl)propanamido)ethyl)carbamate (**10f**).

The title compound **10i** was prepared from compound **6a** (0.3 g, 0.55 mmol) and **7f** (0.66 mmol; **7f** in situ preparation from **7f’** using procedure F) according to the general procedure G. The product **10f** was obtained as an off-white solid (294 mg, 77%); 1H NMR (400 MHz, DMSO-*d*_6_) δ 12.26 (s, 1H), 11.88 (s, 1H), 10.80 (s, 1H), 9.14 (d, J = 8.0 Hz, 1H), 8.56 (s, 1H), 8.30 (t, J = 5.5 Hz, 1H), 8.08 (d, J = 2.4 Hz, 1H), 7.73 (dd, J = 2.0, 6.5 Hz, 2H), 7.67 (d, J = 8.0 Hz, 1H), 7.49–7.42 (m, 1H), 7.30–7.16 (m, 3H), 7.12–6.95 (m, 3H), 6.89 (s, 1H), 6.80 (t, J = 5.9 Hz, 1H), 4.81–4.71 (m, 1H), 3.28 (d, J = 4.5 Hz, 1H), 3.22–3.11 (m, 3H), 3.08–3.00 (m, 2H), 1.34 (s, 9H);13C NMR (100 MHz, DMSO-*d*_6_) 171.3, 167.8, 165.0, 156.1, 138.9, 136.5, 135.3, 134.9, 132.6, 132.0, 129.5, 129.1, 128.6, 128.4, 128.1, 127.6, 127.5, 124.0, 123.6, 122.9, 122.8, 121.3, 118.9, 118.7, 115.0, 111.8, 110.9, 77.9, 55.2, 37.8, 36.8, 29.9, 28.6, 27.7; HRMS (ESI): *m*/*z* calcd for C_35_H_35_BrN_6_O_5_ [M + Na]+: 709.1745; found: 709.1749.

tert-butyl (S)-(2-(2-(5-bromo-2-(thiophene-2-carboxamido)benzamido)-3-(1H-indol-3-yl)propanamido)ethyl)carbamate (**10g**).

The title compound **10g** was prepared from compound **6a** (0.3 g, 0.55 mmol) and **7g** (0.66 mmol; **7g** in situ preparation from **7g’** using procedure F) according to the general procedure G. The product **10g** was obtained as an off-white solid (248 mg, 69%); ^1^H NMR (400 MHz, DMSO-*d_6_*) δ 12.10 (s, 1H), 10.78 (s, 1H), 9.11 (d, J = 8.3 Hz, 1H), 8.40 (d, J = 8.9 Hz, 1H), 8.24 (t, J = 5.6 Hz, 1H), 8.03 (d, J = 2.4 Hz, 1H), 7.89 (dd, J = 1.1, 5.0 Hz, 1H), 7.76–7.66 (m, 2H), 7.51 (dd, J = 1.2, 3.8 Hz, 1H), 7.31–7.23 (m, 1H), 7.23–7.15 (m, 2H), 7.07–6.93 (m, 2H), 6.78 (t, J = 5.6 Hz, 1H), 4.76–4.68 (m, 1H), 3.32–3.26 (m, 2H), 3.13 (td, J = 6.0, 10.2 Hz, 3H), 3.06–2.97 (m, 2H), 1.34 (s, 9H);^13^C NMR (100 MHz, DMSO-*d_6_*) 171.5, 167.7, 159.7, 156.1, 138.6, 136.5, 135.3, 133.0, 131.5, 129.1, 128.8, 127.6, 124.0, 122.4, 121.3, 118.9, 118.6, 114.9, 111.8, 110.9, 78.1, 55.0, 28.6, 27.8; HRMS (ESI): *m*/*z* calcd for C_30_H_32_BrN_5_O_5_S [M + Na]+: 676.1200; found: 676.1198.

tert-butyl (S)-(2-(2-(5-bromo-2-(thiophene-3-carboxamido)benzamido)-3-(1H-indol-3-yl)propanamido)ethyl)carbamate (**10h**).

The title compound **10h** was prepared from compound **11a** (0.3 g, 0.55 mmol) and **7h** (0.66 mmol; **7h** in situ preparation from **7h’** using procedure F) according to the general procedure G. The product **10l** was obtained as an off-white solid (252 mg, 70%); ^1^H NMR (400 MHz, DMSO-*d*_6_) δ 11.89 (s, 1H), 10.78 (s, 1H), 9.09 (d, J = 8.3 Hz, 1H), 8.44 (d, J = 8.9 Hz, 1H), 8.23 (t, J = 5.6 Hz, 1H), 8.06 (dd, J = 1.5, 3.0 Hz, 1H), 8.00 (s, 1H), 7.75–7.65 (m, 3H), 7.39 (dd, J = 1.4, 5.1 Hz, 1H), 7.27 (d, J = 8.0 Hz, 1H), 7.18 (d, J = 2.3 Hz, 1H), 7.08–6.93 (m, 2H), 6.79 (t, J = 5.7 Hz, 1H), 4.76–4.67 (m, 1H), 3.30–3.25 (m, 1H), 3.18–3.07 (m, 3H), 3.05–2.98 (m, 2H), 1.35 (s, 9H);13C NMR (100 MHz, DMSO-*d*_6_) 171.5, 167.7, 160.6, 156.1, 138.8, 138.0, 136.5, 135.2, 131.5, 130.5, 128.5, 127.6, 126.3, 124.0, 122.6, 122.5, 121.3, 118.9, 118.6, 114.8, 111.8, 110.9, 78.1, 55.0, 28.6, 27.7; HRMS (ESI): *m*/*z* calcd for C_30_H_32_BrN_5_O_5_S [M + Na]+: 676.1200; found: 676.1205.

tert-butyl (S)-(2-(2-(2-([1,1′-biphenyl]-2-carboxamido)-5-bromobenzamido)-3-(1H-indol-3-yl)propanamido)ethyl)carbamate (**10i**).

The title compound **10i** was prepared from compound **11a** (0.3 g, 0.55 mmol) and **7i** (0.66 mmol; **7i** in situ preparation from **7i’** using procedure F) according to the general procedure G. The product **10i** was obtained as an off-white solid (271 mg, 68%); ^1^H NMR (400 MHz, DMSO-*d*_6_) δ 11.28 (s, 1H), 10.81 (s, 1H), 8.89 (d, J = 7.9 Hz, 1H), 8.21 (d, J = 8.8 Hz, 1H), 8.14 (t, J = 5.7 Hz, 1H), 7.91 (s, 1H), 7.73–7.34 (m, 7H), 7.35–7.29 (m, 3H), 7.29–7.13 (m, 4H), 7.06 (t, J = 7.6 Hz, 1H), 7.02–6.94 (m, 1H), 6.75 (t, J = 5.8 Hz, 1H), 4.60–4.50 (m, 1H), 3.27–3.18 (m, 1H), 3.18–2.99 (m, 3H), 2.95 (t, J = 6.4 Hz, 2H), 1.35 (s, 9H);^13^C NMR (101 MHz, DMSO-*d*_6_) 171.5, 167.7, 167.0, 156.0, 140.1, 140.0, 138.3, 136.7, 136.5, 134.9, 131.3, 130.88, 130.8, 128.7, 128.6, 127.9, 127.7, 127.6, 124.0, 123.2, 122.6, 121.3, 118.8, 118.7, 115.1, 111.8, 110.8, 78.1, 55.0, 28.6, 27.7; HRMS (ESI): *m*/*z* calcd for C_38_H_38_BrN_5_O_5_ [M + Na]+: 746.1949; found: 746.1953.

tert-butyl (S)-(2-(2-(2-([1,1′-biphenyl]-3-carboxamido)-5-bromobenzamido)-3-(1H-indol-3-yl)propanamido)ethyl)carbamate (**10j**).

The title compound **10j** was prepared from compound **11a** (0.3 g, 0.55 mmol) and **7j** (0.66 mmol; **7j** in situ preparation from **7j’** using procedure F) according to the general procedure G. The product **10j** was obtained as an off-white solid (258 mg, 65%); ^1^H NMR (400 MHz, DMSO-*d*_6_) δ 12.15 (s, 1H), 10.79 (s, 1H), 9.11 (d, J = 8.1 Hz, 1H), 8.52 (d, J = 9.0 Hz, 1H), 8.22 (t, J = 5.9 Hz, 1H), 8.11 (s, 1H), 8.01 (d, J = 2.3 Hz, 1H), 7.91 (dt, J = 1.4, 7.6 Hz, 1H), 7.82–7.65 (m, 5H), 7.62 (t, J = 7.8 Hz, 1H), 7.55–7.46 (m, 2H), 7.46–7.38 (m, 1H), 7.26 (d, J = 8.0 Hz, 1H), 7.19 (s, 1H), 7.06–6.98 (m, 1H), 6.98–6.90 (m, 1H), 6.80–6.72 (m, 1H), 4.75–4.65 (m, 1H), 3.31–3.25 (m, 1H), 3.19–3.02 (m, 3H), 3.02–2.93 (m, 2H), 1.33 (s, 9H);13C δ C (101 MHz, DMSO-*d_6_*) 171.5, 167.7, 164.9, 156.1, 141.2, 139.7, 138.8, 136.5, 135.5, 135.2, 131.6, 130.8, 130.0, 129.6, 129.3, 128.4, 127.6, 127.2, 126.2, 125.8, 124.0, 123.0, 122.7, 121.3, 118.8, 118.6, 115.11, 111.8, 110.9, 78.1, 55.1, 28.6, 27.7; HRMS (ESI): *m*/*z* calcd for C_38_H_38_BrN_5_O_5_ [M + Na]+: 746.1949; found: 746.1953.

tert-butyl (S)-(2-(2-(2-([1,1′-biphenyl]-4-carboxamido)-5-bromobenzamido)-3-(1H-indol-3-yl)propanamido)ethyl)carbamate (**10k**).

The title compound **10k** was prepared from compound **6a** (0.3 g, 0.55 mmol) and **7k** (0.66 mmol; **7k** in situ preparation from **7k’** using procedure F) according to the general procedure G. The product **10k** was obtained as an off-white solid (282 mg, 71%); ^1^H NMR (400 MHz, DMSO-*d*_6_) δ 12.13 (s, 1H), 10.80 (s, 1H), 9.14 (d, J = 8.1 Hz, 1H), 8.56 (d, J = 9.0 Hz, 1H), 8.27 (t, J = 5.8 Hz, 1H), 8.03 (s, 1H), 7.92–7.84 (m, 2H), 7.82 (d, J = 8.6 Hz, 2H), 7.78–7.67 (m, 4H), 7.51 (dd, J = 6.7, 8.3 Hz, 2H), 7.47–7.39 (m, 1H), 7.31–7.24 (m, 1H), 7.20 (d, J = 2.4 Hz, 1H), 7.06–6.93 (m, 2H), 6.80 (t, J = 5.8 Hz, 1H), 4.77–4.66 (m, 1H), 3.31–3.26 (m, 1H), 3.20–3.09 (m, 3H), 3.02 (q, J = 6.5 Hz, 2H), 1.32 (s, 9H);^13^C NMR (100 MHz, DMSO-*d*_6_) 171.15, 167.8, 164.6, 156.1, 144.1, 139.3, 138.9, 136.5, 135.3, 132.9, 131.1, 129.1, 128.3, 127.6, 127.2, 127.1, 127.0, 123.6, 122.2, 122.1, 120.9, 118.5, 118.2, 114.5, 111.3, 110.5, 77.7, 54.7, 28.2, 27.3; HRMS (ESI): *m*/*z* calcd for C_38_H_38_BrN_5_O_5_ [M + Na]+: 746.1949; found: 746.1954.

tert-butyl (S)-(3-(2-(2-(2-naphthamido)-5-bromobenzamido)-3-(1H-indol-3-yl)propanamido)propyl)carbamate (**11a**).

The title compound **11a** was prepared from compound **6b** (0.3 g, 0.54 mmol) and **7a** (0.66 mmol; **7a** in situ preparation from **7a’** using procedure F) according to the general procedure G. The product **11a** was obtained as an off-white solid (230 mg, 60%); ^1^H NMR (400 MHz, DMSO-*d*_6_) δ 12.23 (s, 1H), 10.80 (s, 1H), 9.17 (d, J = 8.0 Hz, 1H), 8.56 (d, J = 8.9 Hz, 1H), 8.43 (d, J = 2.0 Hz, 1H), 8.18 (t, J = 5.8 Hz, 1H), 8.09–7.98 (m, 4H), 7.84 (dd, J = 2.0, 8.7 Hz, 1H), 7.76 (dd, J = 2.3, 9.0 Hz, 1H), 7.73–7.58 (m, 3H), 7.30–7.23 (m, 1H), 7.21 (s, 1H), 7.05–6.93 (m, 2H), 6.72 (t, J = 5.8 Hz, 1H), 4.77–4.68 (m, 1H), 3.30–3.25 (m, 1H), 3.21–3.04 (m, 3H), 2.90 (q, J = 6.2, 6.7 Hz, 2H), 1.50 (p, J = 6.8 Hz, 2H), 1.32 (s, 9H); ^13^C NMR (100 MHz, DMSO-*d*_6_) 171.3, 167.8, 165.0, 156.0, 138.9, 136.5, 135.2, 134.9, 132.6, 132.0, 131.5, 129.5, 129.1, 128.6, 128.4, 128.1, 127.6, 127.5, 124.0, 123.6, 122.9, 122.8, 121.3, 118.9, 118.7, 115.0, 111.8, 110.9, 77.9, 55.2, 37.8, 36.8, 29.9, 28.6, 27.7; HRMS (ESI): *m*/*z* calcd for C_37_H_38_BrN_5_O_5_ [M + Na]+: 734.1949; found: 734.1952.

tert-butyl (S)-(3-(2-(5-bromo-2-(2-(naphthalen-1-yl)acetamido)benzamido)-3-(1H-indol-3-yl)propanamido)propyl)carbamate (**11b**).

The title compound **11b** was prepared from compound **6b** (0.3 g, 0.54 mmol) and **7b** (0.66 mmol; **7b** in situ preparation from **7b’** using procedure F) according to the general procedure G. The product **11b** was obtained as an off-white solid (263 mg, 67%); ^1^H NMR (400 MHz, DMSO-*d*_6_) δ 10.97 (s, 1H), 10.82 (s, 1H), 8.90 (d, J = 8.0 Hz, 1H), 8.19 (d, J = 8.9 Hz, 1H), 8.10 (t, J = 5.8 Hz, 1H), 8.05–7.97 (m, 1H), 7.95–7.87 (m, 1H), 7.87–7.77 (m, 2H), 7.69 (d, J = 7.5 Hz, 1H), 7.60 (dd, J = 2.4, 8.9 Hz, 1H), 7.54–7.30 (m, 5H), 7.16 (s, 1H), 7.11–7.04 (m, 1H), 7.04–6.96 (m, 1H), 6.76 (t, J = 5.8 Hz, 1H), 4.81–4.48 (m, 1H), 4.10 (d, J = 8.5 Hz, 2H), 3.29–3.20 (m, 1H), 3.14–3.00 (m, 3H), 2.97–2.87 (m, 2H), 1.50 (p, J = 6.3 Hz, 2H), 1.35 (s, 9H);^13^C NMR δ (101 MHz, DMSO-*d*_6_) 171.5, 169.7, 167.0, 156.1, 137.9, 136.5, 134.6, 133.8, 132.3, 131.8, 131.2, 128.9, 128.7, 128.0, 127.7, 126.7, 126.2, 126.0, 124.4, 124.0, 1202.8, 121.4, 118.9, 118.7, 114.8, 111.8, 110.9, 77.9, 60.2, 54.9, 42.2, 37.8, 36.8, 29.5, 28.7; HRMS (ESI): *m*/*z* calcd for C_38_H_40_BrN_5_O_5_ [M + Na]+: 748.2105; found: 748.2105.

tert-butyl (S)-(3-(2-(5-bromo-2-(2-methoxy-1-naphthamido)benzamido)-3-(1H-indol-3-yl)propanamido)propyl)carbamate (**11c**).

The title compound **11c** was prepared from compound **6b** (0.3 g, 0.54 mmol) and **7c** (0.66 mmol; **7c** in situ preparation from **7c’** using procedure F) according to the general procedure G. The product **11c** was obtained as an off-white solid (280 mg, 70%); ^1^H NMR (400 MHz, DMSO-*d_6_*) δ 11.53 (s, 1H), 10.79 (s, 1H), 9.02 (d, J = 8.0 Hz, 1H), 8.60 (d, J = 9.0 Hz, 1H), 8.10–7.97 (m, 3H), 7.92 (dd, J = 1.5, 8.1 Hz, 1H), 7.83–7.73 (m, 2H), 7.63 (d, J = 7.7 Hz, 1H), 7.54–7.43 (m, 2H), 7.43–7.35 (m, 1H), 7.29 (d, J = 8.1 Hz, 1H), 7.16 (s, 1H), 7.08–6.99 (m, 1H), 6.99–6.91 (m, 1H), 4.63–4.45 (m, 1H), 3.78 (s, 3H), 3.20 (dd, J = 4.4, 14.3 Hz, 1H), 3.09 (dd, J = 10.1, 14.6 Hz, 1H), 3.05–2.89 (m, 2H), 2.87–2.66 (m, 2H), 1.38 (s, 1H), 1.43–1.37 (m, 2H), 1.35 (s, 9H);^13^C NMR (101 MHz, DMSO-*d*_6_) 171.6, 170.8, 167.2, 165.4, 156.0, 153.8, 138.6, 136.5, 135.2, 131.9, 131.5, 131.0, 128.66, 128.6, 128.0, 127.6, 124.5, 124.0, 122.9, 122.7, 121.3, 120.4, 118.8, 118.6, 115.0, 114.2, 111.7, 110.8, 78.1, 60.2, 56.8, 54.9, 37.3, 36.2, 28.6, 27.7, 21.2; HRMS (ESI): *m*/*z* calcd for C_38_H_40_BrN_5_O_6_ [M + Na]+: 764.2057; found: 764.2054.

tert-butyl (S)-(3-(2-(5-bromo-2-(3-methoxy-2-naphthamido)benzamido)-3-(1H-indol-3-yl)propanamido)propyl)carbamate (**11d**).

The title compound **11d** was prepared from compound **6b** (0.3 g, 0.54 mmol) and **7d** (0.66 mmol; **7d** in situ preparation from **7d’** using procedure F) according to the general procedure G. The product **11d** was obtained as an off-white solid (270 mg, 65%); 1H NMR (400 MHz, DMSO-*d*_6_) δ 11.86 (s, 1H), 10.79 (s, 1H), 8.99 (d, J = 8.2 Hz, 1H), 8.65 (d, J = 9.1 Hz, 1H), 8.57 (s, 1H), 8.13 (t, J = 5.8 Hz, 1H), 8.00 (d, J = 8.1 Hz, 1H), 7.88 (d, J = 8.3 Hz, 1H), 7.82 (d, J = 2.5 Hz, 1H), 7.75–7.67 (m, 2H), 7.58 (t, J = 7.6 Hz, 1H), 7.46 (s, 1H), 7.42 (t, J = 7.6 Hz, 1H), 7.26–7.16 (m, 2H), 7.05–6.92 (m, 2H), 6.78–6.70 (m, 1H), 4.69 (td, J = 4.4, 8.6, 9.2 Hz, 1H), 3.88 (s, 3H), 3.30–3.21 (m, 1H), 3.18–3.01 (m, 3H), 2.93–2.84 (m, 2H), 1.48 (p, J = 7.4 Hz, 2H), 1.35 (s, 9H);13C NMR (100 MHz, DMSO-*d*_6_) 171.6, 167.0, 163.5, 156.0, 154.5, 137.9, 136.5, 136.1,134.5, 133.3, 131.5, 129.3, 128.9, 127.9, 127.6, 125.2, 124.9, 124.0, 123.7, 123.5, 121.3, 118.8, 118.6, 114.8, 111.1, 110.9, 107.3, 77.9, 65.3, 55.9, 55.1, 37.4, 36.3, 29.9, 28.7, 27.9; HRMS (ESI): *m*/*z* calcd for C_38_H_40_BrN_5_O_6_ [M + Na]+: 764.2054; found: 764.2059.

tert-butyl (S)-(3-(2-(5-bromo-2-(quinoline-2-carboxamido)benzamido)-3-(1H-indol-3-yl)propanamido)propyl)carbamate (**11e**).

The title compound **11e** was prepared from compound **11e** (0.3 g, 0.54 mmol) and **7e** (0.66 mmol; **7e** in situ preparation from **7e’** using procedure F) according to the general procedure G. The product **11e** was obtained as an off-white solid (265 mg, 69%); 1H NMR (400 MHz, DMSO-*d*_6_) δ 12.95 (s, 1H), 10.76 (s, 1H), 9.03 (d, J = 7.9 Hz, 1H), 8.70 (d, J = 9.0 Hz, 1H), 8.63 (d, J = 8.5 Hz, 1H), 8.24 (d, J = 8.5 Hz, 1H), 8.17 (t, J = 5.8 Hz, 1H), 8.10 (dd, J = 8.2, 14.7 Hz, 2H), 7.99 (s, 1H), 7.90 (t, J = 7.2 Hz, 1H), 7.80–7.70 (m, 3H), 7.28–7.20 (m, 2H), 6.97 (dt, J = 6.8, 20.7 Hz, 2H), 6.68 (t, J = 5.5 Hz, 1H), 4.78 (td, J = 5.2, 9.7 Hz, 1H), 3.30–3.24 (m, 1H), 3.24–3.14 (m, 1H), 3.13–3.02 (m, 2H), 2.89–2.80 (m, 2H), 1.46 (q, J = 7.0 Hz, 2H), 1.33 (s, 9H);13C NMR (100 MHz, DMSO-*d*_6_) 171.5, 167.1, 163.1, 156.0, 149.8, 146.2, 138.7, 137.9, 136.5, 135.0, 131.7, 131.1, 129.8, 128.9, 128.5, 127.7, 124.0, 123.9, 122.3, 121.3, 119.0, 118.9, 118.9, 118.6, 115.1, 111.7, 110.9, 77.9, 55.1, 37.2, 36.7, 29.9, 28.6, 27.8; HRMS (ESI): *m*/*z* calcd for C_36_H_37_BrN_6_O_5_ [M + Na]+: 735.1901; found: 735.1905.

The analytical data for **12a** were already published [20].

(S)-N-(1-((2-aminoethyl)amino)-3-(1H-indol-3-yl)-1-oxopropan-2-yl)-5-bromo-2-(2-(naphthalen-1-yl)acetamido)benzamide (**12b**).

The title compound **12b** was prepared from compound **10c** (0.1 g, 0.14 mmol) according to the general procedure H. The product **12b** was obtained as gummy solid (0.053 g, 64%); 1H NMR (400 MHz, DMSO-*d*_6_) δ 10.86 (s, 2H), 8.91 (d, J = 8.0 Hz, 1H), 8.29 (t, J = 5.8 Hz, 1H), 8.18 (d, J = 9.0 Hz, 1H), 8.03–7.96 (m, 1H), 7.91 (dt, J = 3.0, 8.6 Hz, 1H), 7.88–7.77 (m, 5H), 7.66 (d, J = 7.8 Hz, 1H), 7.61 (dd, J = 2.4, 8.9 Hz, 1H), 7.55–7.46 (m, 2H), 7.43 (t, J = 7.6 Hz, 1H), 7.39–7.32 (m, 2H), 7.14 (d, J = 2.4 Hz, 1H), 7.08 (t, J = 7.0 Hz, 1H), 7.00 (t, J = 7.4 Hz, 1H), 4.70–4.60 (m, 1H), 4.18–3.99 (m, 2H), 3.32–3.27 (m, 2H), 3.19–3.03 (m, 2H), 2.87–2.78 (m, 2H);13C NMR δ C (101 MHz, DMSO-*d*_6_) 172.3, 169.8, 167.1, 137.8, 136.5, 134.7, 133.8, 132.3, 132.3, 131.7, 131.2, 128.9, 128.7, 128.0, 127.7, 126.7, 126.2, 124.4, 124.3, 124.1, 122.9, 121.4, 118.9, 118.7, 114.9, 111.8, 110.7, 54.8, 46.1, 42.2, 38.4, 36.6, 27.6; HRMS (ESI): *m*/*z* calcd for C_32_H_30_BrN_5_O_3_ [M + H]+: 612.1605; found: 612.1606.

(S)-N-(2-((1-((2-aminoethyl)amino)-3-(1H-indol-3-yl)-1-oxopropan-2-yl)carbamoyl)-4-bromophenyl)-2-methoxy-1-naphthamide (**12c**).

The title compound **12c** was prepared from compound **10c** (0.1 g, 0.137 mmol) according to the general procedure H. The product **12c** was obtained as off-white solid (0.051 g, 60%); 1H NMR (600 MHz, DMSO-*d*_6_) δ 10.82 (s, 1H), 9.13 (d, J = 8.1 Hz, 1H), 8.58 (d, J = 8.9 Hz, 1H), 8.28 (t, J = 5.6 Hz, 1H), 8.06 (d, J = 9.2 Hz, 1H), 7.99 (s, 1H), 7.92 (dd, J = 1.3, 8.2 Hz, 1H), 7.80 (d, J = 8.5 Hz, 1H), 7.77 (dd, J = 2.4, 8.9 Hz, 1H), 7.63 (d, J = 7.8 Hz, 1H), 7.54–7.45 (m, 2H), 7.43–7.36 (m, 1H), 7.29 (d, J = 8.0 Hz, 1H), 7.17 (s, 1H), 7.06–7.00 (m, 1H), 6.98–6.91 (m, 1H), 4.60–4.54 (m, 1H), 3.79 (s, 3H), 3.24–3.19 (m, 1H), 3.12–3.06 (m, 3H), 2.57 (t, J = 6.8 Hz, 2H);13C NMR (150 MHz, DMSO-*d*_6_) 171.7, 167.2, 166.4, 165.4, 153.8, 138.5, 136.4, 135.1, 131.9, 131.5, 131.0, 128.6, 128.60, 128.0, 127.6, 124.5, 124.0, 123.2, 122.8, 121.3, 120.4, 118.8, 118.7, 115.1, 114.3, 111.8, 110.8, 56.8, 55.0, 27.2; HRMS (ESI): *m*/*z* calcd for C_32_H_30_BrN_5_O_4_ [M + H]+: 628.1554; found: 628.1557.

(S)-N-(2-((1-((2-aminoethyl)amino)-3-(1H-indol-3-yl)-1-oxopropan-2-yl)carbamoyl)-4-bromophenyl)-3-methoxy-2-naphthamide (**12d**).

The title compound **12d** was prepared from compound **10d** (0.1 g, 0.137 mmol) according to the general procedure H. The product **12d** was obtained as off-white solid (0.054 g, 63%); 1H NMR (600 MHz, DMSO-*d*_6_) δ 10.80 (s, 1H), 9.09 (s, 1H), 8.63 (d, J = 8.9 Hz, 1H), 8.57 (s, 1H), 8.43 (s, 2H), 8.00 (d, J = 8.0 Hz, 1H), 7.88 (d, J = 8.0 Hz, 1H), 7.84 (d, J = 2.5 Hz, 1H), 7.73–7.67 (m, 2H), 7.58 (ddd, J = 1.4, 6.8, 8.2 Hz, 1H), 7.48 (s, 1H), 7.43 (ddd, J = 1.4, 6.8, 8.1 Hz, 1H), 7.22 (d, J = 8.0 Hz, 1H), 7.18 (d, J = 2.5 Hz, 1H), 7.03–6.92 (m, 2H), 4.75–4.68 (m, 1H), 3.89 (s, 3H), 3.30–3.07 (m, 6H), 2.67 (td, J = 2.6, 6.4 Hz, 2H); HRMS (ESI): *m*/*z* calcd for C_32_H_30_BrN_5_O_4_ [M + H]+: 628.1554; found: 628.1555.

(S)-N-(2-((1-((2-aminoethyl)amino)-3-(1H-indol-3-yl)-1-oxopropan-2-yl)carbamoyl)-4-bromophenyl)quinoline-2-carboxamide (**12e**).

The title compound **12e** was prepared from compound **10h** (0.1 g, 0.14 mmol) according to the general procedure H. The product **12e** was obtained as gummy liquid (0.055 g, 64%); 1H NMR (600 MHz, DMSO-*d*_6_) δ 12.93 (s, 1H), 10.76 (s, 1H), 9.05 (d, J = 7.8 Hz, 1H), 8.70 (d, J = 8.9 Hz, 1H), 8.65 (d, J = 8.5 Hz, 1H), 8.37 (t, J = 5.8 Hz, 1H), 8.26 (d, J = 8.5 Hz, 1H), 8.13 (dd, J = 1.6, 8.2 Hz, 1H), 8.12–8.05 (m, 1H), 8.00 (d, J = 2.5 Hz, 1H), 7.94–7.87 (m, 1H), 7.82–7.75 (m, 3H), 7.75 (s, 2H), 7.70 (d, J = 7.8 Hz, 1H), 7.28–7.18 (m, 2H), 6.98 (ddd, J = 1.3, 6.9, 8.2 Hz, 1H), 6.92 (td, J = 1.1, 7.0, 7.5 Hz, 1H), 4.87–4.79 (m, 1H), 3.31–3.17 (m, 3H), 2.83 (t, J = 6.8 Hz, 2H);13C NMR (150 MHz, DMSO-*d*_6_) 1172.2, 167.1, 163.1, 158.1, 149.8, 146.2, 138.8, 137.9, 136.5, 135.1, 131.8, 131.2, 129.8, 128.6, 127.7, 124.0, 123.9, 122.4, 121.3, 119.0, 118.8, 118.6, 115.1, 111.7, 110.7, 54.9, 38.4, 36.6, 27.5; HRMS (ESI): *m*/*z* calcd for C_30_H_27_BrN_6_O_3_ [M + H]+: 599.1401; found: 599.1403.

(S)-N-(2-((1-((2-aminoethyl)amino)-3-(1H-indol-3-yl)-1-oxopropan-2-yl)carbamoyl)-4-bromophenyl)-1H-indole-2-carboxamide (**12f**).

The title compound **12f** was prepared from compound **10f** (0.1 g, 0.15 mmol) according to the general procedure H. The product **12f** was obtained as gummy liquid (0.057 g, 67%); 1H NMR (400 MHz, DMSO-*d*_6_) δ 12.21 (s, 1H), 11.90 (s, 1H), 10.82 (s, 1H), 9.17 (d, J = 8.1 Hz, 1H), 8.56 (d, J = 8.9 Hz, 1H), 8.41 (t, J = 5.8 Hz, 1H), 8.08 (s, 1H), 7.82 (s, 2H), 7.80–7.60 (m, 4H), 7.47 (d, J = 8.3 Hz, 1H), 7.33–7.16 (m, 3H), 7.15–6.95 (m, 3H), 6.89 (s, 1H), 4.86–4.76 (m, 1H), 3.37 (dd, J = 4.3, 8.5 Hz, 3H), 3.23–3.12 (m, 1H), 2.86 (t, J = 6.8 Hz, 2H);13C NMR (100 MHz, DMSO-*d*_6_) 172.1, 167.8, 159.6, 158.3, 139.0, 137.6, 136.5, 135.5, 131.8, 131.6, 127.6, 127.3, 124.6, 124.0, 122.3, 122.2, 121.7, 121.4, 120.7, 118.7, 114.6, 113.0, 111.8, 110.7, 103.3, 54.9, 39.9, 37.1, 27.6; HRMS (ESI): *m*/*z* calcd for C_29_H_27_BrN_6_O_3_ [M + H]+: 587.1401; found: 587.1407.

(S)-N-(2-((1-((2-aminoethyl)amino)-3-(1H-indol-3-yl)-1-oxopropan-2-yl)carbamoyl)-4-bromophenyl)thiophene-2-carboxamide (**12g**).

The title compound **12g** was prepared from compound **10g** (0.1 g, 0.15 mmol) according to the general procedure H. The product **12g** was obtained as off-white solid (0.058 g, 69%); 1H NMR (400 MHz, DMSO-*d*_6_) δ 12.01 (s, 1H), 10.81 (s, 1H), 9.14 (d, J = 8.0 Hz, 1H), 8.45–8.31 (m, 2H), 8.01 (s, 1H), 7.91 (dd, J = 1.2, 5.0 Hz, 1H), 7.83 (s, 2H), 7.73 (dd, J = 2.3, 8.9 Hz, 1H), 7.67 (d, J = 7.5 Hz, 1H), 7.52 (dd, J = 1.2, 3.8 Hz, 1H), 7.27 (d, J = 7.9 Hz, 1H), 7.24–7.15 (m, 2H), 7.07–6.93 (m, 2H), 4.80–4.70 (m, 1H), 3.39–3.33 (m, 5H), 3.20–3.09 (m, 1H), 2.85 (t, J = 6.9 Hz, 2H);13C NMR (100 MHz, DMSO-*d*_6_) 1172.1, 167.7, 159.7, 139.8, 138.5, 136.5, 135.3, 133.0, 131.5, 129.1, 128.9, 127.6, 124.0, 122.7, 121.4, 118.8, 118.7, 115.0, 111.8, 110.7, 54.9, 339.9, 38.8, 27.5; HRMS (ESI): *m*/*z* calcd for C_25_H_24_BrN_5_O_3_S [M + H]+: 554.0856; found: 554.0860.

(S)-N-(2-((1-((2-aminoethyl)amino)-3-(1H-indol-3-yl)-1-oxopropan-2-yl)carbamoyl)-4-bromophenyl)thiophene-3-carboxamide (**12h**).

The title compound **12h** was prepared from compound **10h** (0.1 g, 0.15 mmol) according to the general procedure H. The product **12h** was obtained as off-white solid (0.055 g, 67%); 1H NMR (400 MHz, DMSO-*d*_6_) δ 11.83 (s, 1H), 10.81 (s, 1H), 9.12 (d, J = 8.1 Hz, 1H), 8.42 (d, J = 8.9 Hz, 1H), 8.35 (t, J = 5.7 Hz, 1H), 8.06 (dd, J = 1.4, 3.0 Hz, 1H), 7.99 (d, J = 2.3 Hz, 1H), 7.82 (s, 2H), 7.76–7.57 (m, 4H), 7.38 (s, 1H), 7.28 (d, J = 8.0 Hz, 1H), 7.18 (s, 1H), 7.06–6.93 (m, 2H), 4.80–4.70 (m, 1H), 3.51–3.35 (m, 2H), 3.25–3.04 (m, 2H), 2.84 (t, J = 6.8 Hz, 2H);13C NMR (100 MHz, DMSO-*d*_6_) 172.1, 167.7, 160.7, 138.7, 138.0, 136.5, 135.3, 131.5, 130.5, 128.5, 127.6, 126.3, 124.0, 122.7, 121.4, 118.8, 114.9, 111.8, 110.7, 54.9, 39.9, 38.8, 27.5;.; HRMS (ESI): *m*/*z* calcd for C_25_H_24_BrN_5_O_3_S [M + H]+: 554.0856; found: 554.0859.

(S)-N-(2-((1-((2-aminoethyl)amino)-3-(1H-indol-3-yl)-1-oxopropan-2-yl)carbamoyl)-4-bromophenyl)-[1,1′-biphenyl]-2-carboxamide (**12i**).

The title compound **12i** was prepared from compound **10m** (0.1 g, 0.14 mmol) according to the general procedure H. The product **12i** was obtained as gummy solid (0.058 g, 67%); 1H NMR (600 MHz, DMSO-*d*_6_) δ 11.22 (s, 1H), 10.85 (d, J = 2.5 Hz, 1H), 8.93 (d, J = 7.8 Hz, 1H), 8.25 (t, J = 5.7 Hz, 1H), 8.16 (d, J = 8.9 Hz, 1H), 7.89 (d, J = 2.3 Hz, 1H), 7.85–7.71 (m, 3H), 7.70–7.42 (m, 7H), 7.37–7.19 (m, 7H), 7.16 (d, J = 2.5 Hz, 1H), 7.06 (ddd, J = 1.4, 6.9, 8.2 Hz, 1H), 6.98 (td, J = 1.2, 7.0, 7.5 Hz, 1H), 4.62–4.54 (m, 1H), 3.31–3.20 (m, 3H), 3.10 (dd, J = 9.8, 14.8 Hz, 1H), 2.77 (t, J = 6.9 Hz, 2H).; 13C NMR (150 MHz, DMSO-*d*_6_) 172.1, 167.8, 167.1, 158.5, 158.3, 140.1, 140.0, 138.2, 136.7, 136.5, 131.3, 130.9, 130.8, 128.7, 128.0, 127.8, 127.6, 124.0, 123.5, 122.8, 121.4, 118.8, 118.7, 115.2, 111.8, 110.7, 65.3, 54.9, 39.5, 37.0, 27.5; HRMS (ESI): *m*/*z* calcd for C_33_H_30_BrN_5_O_3_ [M + H]+: 624.1605; found: 624.1607.

(S)-N-(2-((1-((2-aminoethyl)amino)-3-(1H-indol-3-yl)-1-oxopropan-2-yl)carbamoyl)-4-bromophenyl)-[1,1′-biphenyl]-3-carboxamide (**12j**).

The title compound **12j** was prepared from compound **10j** (0.1 g, 0.14 mmol) according to the general procedure H. The product **12j** was obtained as off-white solid (0.061 g, 70%); 1H NMR (400 MHz, DMSO-*d*_6_) δ 12.09 (s, 1H), 10.82 (s, 1H), 9.14 (d, J = 8.0 Hz, 1H), 8.50 (d, J = 9.0 Hz, 1H), 8.34 (q, J = 6.6, 7.4 Hz, 1H), 8.12 (s, 1H), 8.01 (s, 1H), 7.92 (dt, J = 1.4, 7.7 Hz, 1H), 7.82–7.68 (m, 6H), 7.68–7.59 (m, 2H), 7.51 (dd, J = 6.7, 8.4 Hz, 2H), 7.47–7.38 (m, 1H), 7.26 (d, J = 8.0 Hz, 1H), 7.18 (s, 1H), 7.06–6.98 (m, 1H), 6.98–6.90 (m, 1H), 4.79–4.69 (m, 1H), 3.36–3.11 (m, 4H), 2.82 (q, J = 5.7, 6.1 Hz, 2H);13C NMR (100 MHz, DMSO-*d*_6_) 1172.1, 167.7, 164.9, 141.2, 139.6, 138.7, 136.5, 135.5, 135.3, 131.6, 130.8, 130.1, 129.6, 128.4, 127.6, 127.3, 126.2, 125.8, 124.0, 118.8, 118.7, 115.2, 111.8, 110.8, 54.9, 42.2, 37.0, 27.5; HRMS (ESI): *m*/*z* calcd for C_33_H_30_BrN_5_O_3_ [M + H]+: 624.1605; found: 624.1606.

(S)-N-(2-((1-((2-aminoethyl)amino)-3-(1H-indol-3-yl)-1-oxopropan-2-yl)carbamoyl)-4-bromophenyl)-[1,1′-biphenyl]-4-carboxamide (**12k**).

The title compound **12k** was prepared from compound **10k** (0.1 g, 0.14 mmol) according to the general procedure H. The product **12k** was obtained as off-white solid (0.060 g, 70%); 1H NMR (600 MHz, DMSO-*d*_6_) δ 12.06 (s, 1H), 10.82 (s, 1H), 9.16 (d, J = 8.1 Hz, 1H), 8.54 (d, J = 8.9 Hz, 1H), 8.38 (t, J = 5.7 Hz, 1H), 8.02 (s, 1H), 7.90–7.73 (m, 11H), 7.92–7.72 (m, 11H), 7.67 (d, J = 7.8 Hz, 1H), 7.52 (t, J = 7.7 Hz, 2H), 7.44 (t, J = 7.4 Hz, 1H), 7.26 (d, J = 8.0 Hz, 1H), 7.19 (s, 1H), 7.00 (dt, J = 7.1, 29.3 Hz, 2H), 4.80–4.74 (m, 1H), 3.41–3.35 (m, 3H), 3.19–3.12 (m, 1H), 2.85 (t, J = 7.2 Hz, 2H);13C NMR (75 MHz, CDCl3): δ δ C (151 MHz, DMSO-*d*_6_) 1172.1, 167.8, 164.6, 158.5, 158.3, 144.1, 139.3, 138.8, 136.5, 135.3, 133.4, 131.5, 129.5, 128.8, 128.1, 127.6, 127.67, 127.6, 127.4, 124.0, 122.8, 122.7, 121.3, 118.8, 118.7, 115.0, 111.8, 110.8, 54.9, 38.8, 37.1, 27.6; HRMS (ESI): *m*/*z* calcd for C_25_H_24_BrN_5_O_3_S [M + H]+: 624.1605; found: 624.1605.

(S)-N-(2-((1-((3-aminopropyl)amino)-3-(1H-indol-3-yl)-1-oxopropan-2-yl)carbamoyl)-4-bromophenyl)-2-naphthamide (**13a**).

The title compound 13a was prepared from compound **11a** (0.1 g, 0.14 mmol) according to the general procedure H. The product 13a was obtained as gummy solid (0.052 g, 62%); 1H NMR (600 MHz, DMSO-*d*_6_) δ 12.18 (s, 1H), 10.82 (s, 1H), 9.20 (d, J = 8.1 Hz, 1H), 8.53 (d, J = 8.9 Hz, 1H), 8.44 (s, 1H), 8.39 (d, J = 6.0 Hz, 1H), 8.08–7.97 (m, 4H), 7.87–7.55 (m, 9H), 7.26 (d, J = 8.1 Hz, 1H), 7.21 (s, 1H), 7.01 (t, J = 7.6 Hz, 1H), 6.97 (t, J = 7.6 Hz, 1H), 4.78–4.68 (m, 1H), 3.29–3.26 (m, 1H), 3.20–3.13 (m, 3H), 2.76–2.70 (m, 2H), 1.68 (p, J = 7.3 Hz, 2H); HRMS (ESI): *m*/*z* calcd for C_32_H_30_BrN_5_O_3_ [M + H]+: 612.1605; found: 612.1606.

(S)-N-(1-((3-aminopropyl)amino)-3-(1H-indol-3-yl)-1-oxopropan-2-yl)-5-bromo-2-(2-(naphthalen-1-yl)acetamido)benzamide (**13b**).

The title compound **13b** was prepared from compound **11b** (0.1 g, 0.14 mmol) according to the general procedure H. The product **13b** was obtained as pale brown solid (0.060 g, 69%); 1H NMR (600 MHz, DMSO-*d*_6_) δ 10.85 (s, 1H), 8.93 (s, 1H), 8.41 (s, 1H), 8.27 (t, J = 5.9 Hz, 1H), 8.17 (d, J = 8.9 Hz, 1H), 8.03–7.98 (m, 1H), 7.94–7.86 (m, 1H), 7.85–7.78 (m, 2H), 7.69 (d, J = 7.8 Hz, 1H), 7.60 (dd, J = 2.3, 8.9 Hz, 1H), 7.53–7.46 (m, 2H), 7.46–7.37 (m, 2H), 7.34 (d, J = 8.1 Hz, 1H), 7.16 (s, 1H), 7.10–7.04 (m, 1H), 7.04–6.98 (m, 1H), 4.64–4.59 (m, 1H), 4.10 (d, J = 15.6 Hz, 2H), 3.20–3.13 (m, 2H), 3.13–3.07 (m, 2H), 2.65 (t, J = 7.1 Hz, 2H), 1.63–1.55 (m, 2H);13C NMR (δ C (151 MHz, DMSO-*d*_6_) 171.9, 169.8, 167.06, 166.0, 137.9, 136.5, 134.5, 133.8, 132.3, 131.8, 131.2, 128.9, 128.7, 128.0, 127.7, 126.7, 126.2, 126.0, 124.4, 124.4, 124.1, 122.9, 121.3, 118.9, 118.7, 114.8, 111.8, 110.8, 54.9, 42.1, 39.5, 27.7; HRMS (ESI): *m*/*z* calcd for C_33_H_32_BrN_5_O_3_ [M + H]+: 626.1761; found: 626.1763.

(S)-N-(2-((1-((3-aminopropyl)amino)-3-(1H-indol-3-yl)-1-oxopropan-2-yl)carbamoyl)-4-bromophenyl)-2-methoxy-1-naphthamide (**13c**).

The title compound **13c** was prepared from compound **11c** (0.1 g, 0.14 mmol) according to the general procedure H. The product **13c** was obtained as off-white solid (0.060 g, 67%); 1H NMR (600 MHz, DMSO-*d*_6_) δ 10.86 (s, 1H), 9.18 (s, 1H), 8.58 (d, J = 8.9 Hz, 1H), 8.47 (s, 1H), 8.28 (t, J = 5.8 Hz, 1H), 8.06 (d, J = 9.1 Hz, 1H), 8.01 (d, J = 2.5 Hz, 1H), 7.94–7.90 (m, 1H), 7.80 (d, J = 8.5 Hz, 1H), 7.77 (dd, J = 2.5, 8.8 Hz, 1H), 7.63 (d, J = 8.0 Hz, 1H), 7.51 (d, J = 9.1 Hz, 1H), 7.47 (ddd, J = 1.4, 6.7, 8.4 Hz, 1H), 7.40 (ddd, J = 1.2, 6.7, 8.1 Hz, 1H), 7.29 (s, 1H), 7.17 (s, 1H), 7.06–7.01 (m, 1H), 6.95 (t, J = 7.4 Hz, 1H), 4.54 (dd, J = 4.9, 10.0 Hz, 1H), 3.77 (s, 3H), 3.23–3.18 (m, 1H), 3.14–2.96 (m, 4H), 2.54 (tt, J = 3.6, 7.3 Hz, 2H), 1.50 (p, J = 7.0 Hz, 2H);13C NMR (150 MHz, DMSO-*d*_6_) 171.9, 167.0, 166.2, 163.5,154.5, 137.9, 136.5, 136.1, 134.5, 133.3, 131.5, 129.3, 128.9, 127.9, 127.7, 126.8, 125.3, 124.9, 124.0, 123.7, 123.5, 121.3, 118.8, 118.6, 114.9, 111.7, 110.8, 107.3, 56.0, 55.1, 40.5, 27.9; HRMS (ESI): *m*/*z* calcd for C_33_H_32_BrN_5_O_4_ [M + H]+: 642.1710; found: 642.1713.

(S)-N-(2-((1-((3-aminopropyl)amino)-3-(1H-indol-3-yl)-1-oxopropan-2-yl)carbamoyl)-4-bromophenyl)-3-methoxy-2-naphthamide (**13d**).

The title compound **13d** was prepared from compound **12d** (0.1 g, 0.14 mmol) according to the general procedure H. The product **13d** was obtained as off-white solid (0.054 g, 60%); 1H NMR (600 MHz, DMSO-*d*_6_) δ 11.85 (s, 1H), 10.85 (s, 1H), 9.15 (d, J = 8.1 Hz, 1H), 8.64 (s, 1H), 8.57 (s, 1H), 8.47–8.41 (m, 2H), 8.00 (d, J = 7.8 Hz, 1H), 7.90–7.81 (m, 2H), 7.73–7.67 (m, 2H), 7.58 (ddd, J = 1.4, 6.8, 8.2 Hz, 1H), 7.47 (s, 1H), 7.42 (ddd, J = 1.2, 6.7, 8.1 Hz, 1H), 7.23 (d, J = 8.0 Hz, 1H), 7.19 (d, J = 2.3 Hz, 1H), 7.04–6.93 (m, 2H), 4.73–4.66 (m, 1H), 3.88 (s, 3H), 3.28–3.24 (m, 1H), 3.19–3.11 (m, 3H), 2.67 (t, J = 7.2 Hz, 2H), 1.66–1.61 (m, 2H).;13C NMR δ C (151 MHz, DMSO-*d*_6_) 172.0-, 167.0, 166.3, 163.5, 154.5, 137.9, 136.5, 136.1, 134.5, 133.3, 131.5, 129.3, 128.9, 127.9, 127.6, 126.8, 125.2, 124.9, 124.1, 123.7, 123.5, 121.3, 118.8, 118.7, 114.9, 111.8, 110.8, 107.3, 56.0, 55.2, 40.4, 37.3, 28.8, 28.0.; HRMS (ESI): *m*/*z* calcd for C_33_H_32_BrN_5_O_4_ [M + H]+: 642.1710; found: 642.1713.

(S)-N-(2-((1-((3-aminopropyl)amino)-3-(1H-indol-3-yl)-1-oxopropan-2-yl)carbamoyl)-4-bromophenyl)quinoline-2-carboxamide (**13e**).

The title compound **13e** was prepared from compound **11e** (0.1 g, 0.14 mmol) according to the general procedure H. The product **13e** was obtained as gummy solid (0.054 g, 63%); 1H NMR (600 MHz, DMSO-*d*_6_) δ 12.93 (s, 1H), 10.77 (s, 1H), 9.07 (d, J = 7.8 Hz, 1H), 8.70 (d, J = 8.9 Hz, 1H), 8.64 (d, J = 8.4 Hz, 1H), 8.39 (t, J = 5.9 Hz, 1H), 8.25 (d, J = 8.4 Hz, 1H), 8.13 (dd, J = 1.5, 8.2 Hz, 1H), 8.10–8.04 (m, 1H), 8.01 (s, 1H), 7.91 (ddd, J = 1.5, 6.9, 8.4 Hz, 1H), 7.82–7.74 (m, 2H), 7.71 (d, J = 7.8 Hz, 1H), 7.66 (s, 2H), 7.28–7.21 (m, 2H), 7.04–6.92 (m, 2H), 4.79 (ddd, J = 5.2, 7.8, 9.8 Hz, 1H), 3.30–3.21 (m, 2H), 3.17 (q, J = 6.6 Hz, 2H), 2.78–2.63 (m, 2H), 1.67 (p, J = 7.0 Hz, 2H);13C NMR δ C (151 MHz, DMSO-*d*_6_) 172.2, 167.2, 163.1, 149.8, 146.2, 138.8, 137.9, 136.5, 135.0, 131.8, 131.1, 129.8, 129.5, 129.0, 128.6, 127.6, 124.0, 123.9, 122.4, 121.3, 119.0, 118.8, 118.6, 115.1, 111.8, 110.8, 55.0, 40.5, 37.1, 36.1, 27.8; HRMS (ESI): *m*/*z* calcd for C_31_H_29_BrN_6_O_3_ [M + H]+: 613.1557; found: 613.1560.

tert-butyl (S)-(2-(2-(4-amino-[1,1′-biphenyl]-3-carboxamido)-3-(1H-indol-3-yl)propanamido)ethyl)carbamate (**15a**).

The title compound **15a** was prepared from compound **5a** (0.25 g, 0.46 mmol) and **16a** (0.084 g, 0.69 mmol) according to the general procedure I. The product **15a** was obtained as an off-white solid (0.151 g, 61%); 1H NMR (300 MHz, DMSO-*d*_6_) δ 10.80 (s, 1H), 8.42 (d, J = 8.0 Hz, 1H), 8.12 (t, J = 5.9 Hz, 1H), 7.86–7.56 (m, 4H), 7.53–7.17 (m, 6H), 7.15–6.89 (m, 3H), 6.85–6.64 (m, 2H), 6.46 (s, 2H), 4.64 (dd, J = 4.9, 8.1 Hz, 1H), 3.30–3.09 (m, 4H), 3.09–2.98 (m, 2H), 1.37 (d, J = 1.5 Hz, 9H);13C NMR (75 MHz, DMSO-*d*_6_) 172.5, 169.0, 156.1, 148.9, 137.6, 136.5, 132.7, 132.5, 132.0, 131.9, 130.3, 129.2, 129.1, 127.8, 126.9, 125.9, 124.0, 121.2, 118.9, 118.6, 117.5, 111.7, 111.2, 78.1, 54.6, 39.3, 34.6, 27.9; HRMS (ESI): *m*/*z* calcd for C_31_H_35_N_5_O_4_ [M + H]+: 542.2762; found: 542.2765.

tert-butyl (S)-(2-(2-(4-amino-4′-(tert-butyl)-[1,1′-biphenyl]-3-carboxamido)-3-(1H-indol-3-yl)propanamido)ethyl)carbamate (**15a**).

The title compound **15a** was prepared from compound **6a** (0.25 g, 0.46 mmol) and **14a** (0.122 g, 0.69 mmol) according to the general procedure I. The product **15a** was obtained as an off-white solid (0.178 g, 65%); 1H NMR (400 MHz, DMSO-*d*_6_) δ 10.81 (s, 1H), 8.41 (d, J = 7.7 Hz, 1H), 8.12 (t, J = 5.9 Hz, 1H), 7.78 (s, 1H), 7.71 (d, J = 7.7 Hz, 1H), 7.69–7.51 (m, 5H), 7.49–7.41 (m, 3H), 7.30 (d, J = 8.1 Hz, 1H), 7.22 (s, 1H), 7.09–7.01 (m, 1H), 7.01–6.93 (m, 1H), 6.76 (d, J = 8.5 Hz, 2H), 4.64 (td, J = 5.0, 9.2 Hz, 1H), 3.18–2.97 (m, 6H), 1.36 (s, 9H), 1.32 (s, 9H);13C NMR (100 MHz, DMSO-*d*_6_) 172.5, 169.0, 156.1, 148.9, 137.6, 136.5, 132.7, 132.5, 132.0, 131.9, 130.3, 129.2, 129.1, 127.8, 126.9, 125.9, 124.0, 121.2, 118.9, 118.6, 117.5, 111.7, 111.2, 78.1, 54.6, 39.3, 34.6, 27.9.

tert-butyl (S)-(2-(2-(2-amino-5-(naphthalen-2-yl)benzamido)-3-(1H-indol-3-yl)propanamido)ethyl)carbamate (**15b**).

The title compound **15b** was prepared from compound **6a** (0.25 g, 0.46 mmol) and **14b** (0.118 g, 0.69 mmol) according to the general procedure I. The product **15b** was obtained as pale yellow solid (0.171 g, 63%); 1H NMR (400 MHz, DMSO-*d*_6_) δ 10.81 (s, 1H), 8.53–8.46 (m, 1H), 8.15 (s, 2H), 8.02–7.85 (m, 5H), 7.76 (d, J = 7.8 Hz, 1H), 7.65 (dd, J = 2.2, 8.6 Hz, 1H), 7.57–7.43 (m, 2H), 7.34–7.23 (m, 2H), 7.09–6.92 (m, 2H), 6.79 (dd, J = 7.1, 9.5 Hz, 2H), 6.53 (s, 2H), 4.67 (td, J = 5.1, 9.3 Hz, 1H), 3.25–2.99 (m, 6H), 1.37 (s, 9H);13C NMR (101 MHz, DMSO-*d*_6_) 172.6, 169.1, 156.1, 149.8, 137.9, 136.5, 133.9, 132.0, 130.7, 128.6, 128.3, 127.9, 127.8, 127.3, 126.7, 126.4, 125.8, 125.2, 124.0, 123.6, 121.2, 119.0, 118.6, 117.4, 114.9, 111.7, 111.3, 78.1, 65.3, 54.7, 39.3, 28.7, 27.9; HRMS (ESI): *m*/*z* calcd for C_35_H_37_N_5_O_4_ [M + Na]+: 614.2738; found: 614.2732.

tert-butyl (S)-(2-(2-(4-amino-4′-fluoro-[1,1′-biphenyl]-3-carboxamido)-3-(1H-indol-3-yl)propanamido)ethyl)carbamate (**15c**).

The title compound **15c** was prepared from compound **6a** (0.25 g, 0.46 mmol) and **14c** (0.096 g, 0.69 mmol) according to the general procedure I. The product **15c** was obtained as an off-white solid (0.167 g, 65%); 1H NMR (400 MHz, DMSO-*d*_6_) δ 10.79 (s, 1H), 8.41 (d, J = 8.3 Hz, 1H), 8.12 (t, J = 5.8 Hz, 1H), 7.93–7.55 (m, 5H), 7.73–7.61 (m, 4H), 7.44 (dd, J = 2.2, 8.5 Hz, 1H), 7.30–7.20 (m, 4H), 7.00 (dt, J = 7.4, 31.7 Hz, 3H), 6.78–6.70 (m, 2H), 6.44 (s, 2H), 4.67–4.61 (m, 1H), 3.20–3.10 (m, 4H), 3.03–2.96 (m, 2H), 1.36 (s, 9H); 13C NMR (100 MHz, DMSO-*d*_6_) 172.5, 169.1, 162.7, 160.3, 156.1, 149.5, 137.0, 136.4, 130.4, 127.9, 127.8, 127.0, 125.7, 124.0, 121.3, 118.9, 118.6, 117.2, 115.9, 115.7, 114.8, 111.7, 111.2, 79.6, 78.1, 54.6, 39.3, 28.6, 27.9; HRMS (ESI): *m*/*z* calcd for C_31_H_34_FN_5_O_4_ [M + Na]+: 582.2487; found: 582.2482.

tert-butyl (S)-(2-(2-(4-amino-4′-(trifluoromethyl)-[1,1′-biphenyl]-3-carboxamido)-3-(1H-indol-3-yl)propanamido)ethyl)carbamate (**15d**).

The title compound **15d** was prepared from compound **5a** (0.25 g, 0.46 mmol) and **15f** (0.131 g, 0.69 mmol) according to the general procedure I. The product **15f** was obtained as an off-white solid (0.168 g, 60%); 1H NMR (400 MHz, DMSO-*d*_6_) δ 10.80 (s, 1H), 8.47 (d, J = 8.3 Hz, 1H), 8.14 (t, J = 5.7 Hz, 1H), 7.89–7.82 (m, 3H), 7.75 (dd, J = 8.1, 19.7 Hz, 3H), 7.56 (dd, J = 2.2, 8.6 Hz, 1H), 7.30 (d, J = 8.0 Hz, 1H), 7.23 (s, 1H), 7.08–7.00 (m, 1H), 7.00–6.92 (m, 1H), 6.82–6.72 (m, 2H), 6.60 (s, 2H), 4.65 (td, J = 4.8, 9.8 Hz, 1H), 3.24–2.97 (m, 6H), 1.36 (s, 9H);13C NMR (100 MHz, DMSO-*d*_6_) 172.5, 168.9, 156.1, 150.4, 144.4, 136.5, 135.1, 130.6, 127.8, 127.7, 126.8, 126.5, 126.4, 126.0, 124.7, 124.1, 123.7, 121.3, 118.9, 118.6, 117.3, 114.9, 118.6, 117.3, 114.9, 111.7, 111.2, 78.1, 54.5, 39.3, 31.4, 28.6, 27.9; HRMS (ESI): *m*/*z* calcd for C_32_H_34_F_3_N_5_O_4_ [M + Na]+: 632.2455; found: 632.2451.

(S)-4-amino-N-(1-((2-aminoethyl)amino)-3-(1H-indol-3-yl)-1-oxopropan-2-yl)-[1,1’-biphenyl]-3-carboxamide (**16a**).

The title compound **16a** was prepared from compound **15a** (0.1 g, 0.18 mmol) according to the general procedure H. The product **16a** was obtained as off-white solid (0.054 g, 67%); 1H NMR (600 MHz, DMSO-*d*_6_) δ 10.86 (s, 1H), 8.84 (d, J = 8.1 Hz, 1H), 8.45 (q, J = 4.7, 5.2 Hz, 1H), 8.15–8.06 (m, 3H), 7.88 (s, 1H), 7.78–7.67 (m, 3H), 7.62 (dd, J = 2.2, 8.4 Hz, 1H), 7.47 (t, J = 7.7 Hz, 3H), 7.36–7.20 (m, 3H), 7.08–7.00 (m, 2H), 6.94 (t, J = 7.4 Hz, 1H), 4.73–4.69 (m, 1H), 3.42–3.31 (m, 3H), 3.25–3.17 (m, 1H), 2.91–2.83 (m, 2H);13C NMR (150 MHz, DMSO-*d*_6_) 172.2, 167.8, 139.4, 136.0, 132.0, 131.5, 130.1, 128.8, 128.7, 127.4, 126.9, 126.8, 126.1, 123.7, 123.6, 120.9, 120.8, 118.5, 118.2, 111.3, 110.7, 110.5, 54.4, 38.4, 36.6, 27.2; HRMS (ESI): *m*/*z* calcd for C_26_H_27_N_5_O_2_ [M + H]+: 442.2237; found: 442.2238.

(S)-4-amino-N-(1-((2-aminoethyl)amino)-3-(1H-indol-3-yl)-1-oxopropan-2-yl)-4′-(tert-butyl)-[1,1′-biphenyl]-3-carboxamide (**16a**).

The title compound **16a** was prepared from compound **15a** (0.1 g, 0.17 mmol) according to the general procedure H. The product **16a** was obtained as off-white solid (0.057 g, 69%); 1H NMR (400 MHz, DMSO-*d*_6_) δ 10.85 (s, 1H), 8.80 (d, J = 8.1 Hz, 1H), 8.42 (t, J = 5.4 Hz, 1H), 8.06 (s, 4H), 7.87 (s, 1H), 7.73 (d, J = 7.6 Hz, 1H), 7.63–7.57 (m, 5H), 7.51–7.44 (m, 3H), 7.30 (d, J = 8.1 Hz, 1H), 7.24 (s, 1H), 7.06–6.93 (m, 4H), 4.71–4.66 (m, 1H), 3.34–3.29 (m, 2H), 3.24–3.15 (m, 2H), 2.88–2.83 (m, 2H), 1.33 (s, 9H);13C NMR (100 MHz, DMSO-*d*_6_) 172.7, 168.7, 149.5, 137.0, 136.5, 130.3, 129.3, 127.8, 127.1, 126.3, 126.0, 124.1, 121.2, 118.9, 118.7, 111.8, 111.1, 54.8, 39.3, 37.0, 27.7, 27.6; HRMS (ESI): *m*/*z* calcd for C_30_H_35_N_5_O_2_ [M + H]+: 498.2864; found: 498.2857.

(S)-2-amino-N-(1-((2-aminoethyl)amino)-3-(1H-indol-3-yl)-1-oxopropan-2-yl)-5-(naphthalen-2-yl)benzamide (**16b**).

The title compound **16b** was prepared from compound **15b** (0.1 g, 0.17 mmol) according to the general procedure H. The product **16b** was obtained as off-white solid (0.058 g, 70%); 1H NMR (400 MHz, DMSO-*d*_6_) δ 10.86 (s, 1H), 8.97 (d, J = 8.1 Hz, 1H), 8.50 (t, J = 5.5 Hz, 1H), 8.31–8.26 (m, 1H), 8.15 (s, 2H), 8.09 (s, 1H), 8.04–7.91 (m, 4H), 7.84–7.75 (m, 2H), 7.60–7.47 (m, 2H), 7.34–7.20 (m, 2H), 7.12 (d, J = 8.4 Hz, 1H), 7.07–6.97 (m, 1H), 6.97–6.88 (m, 1H), 4.78–4.70 (m, 1H), 3.43–3.26 (m, 4H), 2.89 (q, J = 6.1 Hz, 2H);13C NMR δ C (101 MHz, DMSO-*d*_6_) 172.6, 168.1, 137.1, 136.5, 133.8, 132.4, 130.7, 128.8, 128.5, 127.9, 127.8, 127.6, 126.8, 126.3, 125.3, 124.7, 124.1, 121.2, 120.6, 119.0, 118.6, 111.8, 111.2, 66.8, 55.0, 37.0, 27.7; HRMS (ESI): *m*/*z* calcd for C_30_H_35_N_5_O_2_ [M + H]+: 498.2394; found: 498.2390.

(S)-4-amino-N-(1-((2-aminoethyl)amino)-3-(1H-indol-3-yl)-1-oxopropan-2-yl)-4′-fluoro-[1,1′-biphenyl]-3-carboxamide (**16c**).

The title compound **16c** was prepared from compound **15c** (0.1 g, 0.18 mmol) according to the general procedure H. The product **16c** was obtained as off-white solid (0.053 g, 65%); 1H NMR (400 MHz, DMSO-*d*_6_) δ 10.86 (s, 1H), 8.91 (d, J = 8.4 Hz, 1H), 8.47 (t, J = 5.7 Hz, 1H), 8.14 (s, 2H), 7.85 (s, 1H), 7.81–7.65 (m, 3H), 7.62 (dd, J = 2.2, 8.4 Hz, 1H), 7.36–7.20 (m, 4H), 7.11–6.89 (m, 3H), 4.73–4.70 (m, 1H), 3.47–3.29 (m, 3H), 3.27–3.16 (m, 1H), 2.94–2.82 (m, 2H); 13C NMR (100 MHz, DMSO-*d*_6_) 172.6, 168.0, 163.2, 160.8, 136.5, 136.2, 130.4, 128.6, 128.5, 127.8, 127.3, 124.1, 121.2, 120.7, 118.9, 118.7, 116.1, 115.9, 111.8, 111.1, 66.8, 54.8, 40.6, 37.0, 27.7; HRMS (ESI): *m*/*z* calcd for C_26_H_26_FN_5_O_2_ [M + H]+: 460.2143; found: 460.2144.

(S)-4-amino-N-(1-((2-aminoethyl)amino)-3-(1H-indol-3-yl)-1-oxopropan-2-yl)-4′-(trifluoromethyl)-[1,1′-biphenyl]-3-carboxamide (**16d**).

The title compound **16d** was prepared from compound **15d** (0.1 g, 0.16 mmol) according to the general procedure H. The product **16d** was obtained as off-white solid (0.052 g, 62%); 1H NMR (400 MHz, DMSO-*d*_6_) δ 10.85 (s, 1H), 8.76 (d, J = 8.2 Hz, 1H), 8.44 (t, J = 5.7 Hz, 1H), 8.11 (s, 3H), 8.02–7.88 (m, 3H), 7.86–7.57 (m, 5H), 7.46–7.20 (m, 3H), 7.12–6.88 (m, 4H), 4.72–4.68 (m, 1H), 3.42–3.29 (m, 3H), 3.21 (dd, J = 10.0, 14.6 Hz, 1H), 2.87 (h, J = 6.0 Hz, 2H);13C NMR δ C (101 MHz, DMSO-*d*_6_) 172.8, 168.5, 144.1, 136.5, 130.7, 127.89, 127.8, 127.1, 126.7, 126.3, 126.0, 126.05, 124.1, 123.6, 121.2, 118.9, 118.6, 118.68, 111.8, 111.1, 54.7, 40.6, 37.02, 27.8; HRMS (ESI): *m*/*z* calcd for C_27_H_26_F_3_N_5_O_2_ [M + H]+: 510.2108; found: 510.2111.

(S)-2-amino-N-(1-((2-aminoethyl)amino)-3-(1H-indol-3-yl)-1-oxopropan-2-yl)-5-bromobenzamide (**17**).

The title compound **17** was prepared from compound **5a** (0.1 g, 0.17 mmol) according to the general procedure H. The product **17** was obtained as gummy solid (0.064 g, 85%); 1H NMR (400 MHz, DMSO-*d*_6_) δ 10.85 (s, 1H), 8.57 (d, J = 7.9 Hz, 1H), 8.39 (t, J = 5.7 Hz, 1H), 8.12 (s, 2H), 7.77 (d, J = 2.4 Hz, 1H), 7.70 (d, J = 7.6 Hz, 1H), 7.37–7.27 (m, 2H), 7.19 (d, J = 2.4 Hz, 1H), 7.05 (ddd, J = 1.4, 7.0, 8.2 Hz, 1H), 6.98 (ddd, J = 1.2, 6.9, 8.0 Hz, 1H), 6.75 (d, J = 8.8 Hz, 1H), 4.68–4.58 (m, 2H), 3.47–3.03 (m, 4H), 2.90–2.78 (m, 2H);13C NMR (100 MHz, DMSO-*d*_6_) 172.2, 167.1, 146.4, 136.0, 134.4, 130.8, 127.3, 123.6, 120.9, 119.6, 118.5, 118.2, 117.5, 111.4, 110.6, 107.1, 54.3, 38.4, 36.5, 27.2; HRMS (ESI): *m*/*z* calcd for C_20_H_22_BrN_5_O_2_ [M + H]+: 444.1030; found: 444.1037.

di-tert-butyl ((RS)-6-((2-((S)-2-(2-(2-naphthamido)-5-bromobenzamido)-3-(1H-indol-3-yl)propanamido)ethyl)amino)-6-oxohexane-1,5-diyl)dicarbamate (**18a**).

The title compound **18a** was prepared from Boc-Lys(Boc)-Osu (0.142 g, 0.32 mmol) and **12a** (0.2 g, 0.32 mmol) according to the general procedure J. The product **18a** was obtained as an off-white solid (0.16 g, 55%); 1H NMR1H NMR (400 MHz, DMSO-*d*_6_): δ 12.19 (s, 1H), 10.80 (d, J = 1.5 Hz, 1H), 9.14 (d, J = 7.7 Hz, 1H), 8.56 (d, J = 8.9 Hz, 1H), 8.43 (s, 1H), 8.27 (bs, 1H), 8.08–7.99 (m, 4H), 7.91–7.82 (m, 2H), 7.77 (dd, J = 8.9, 2.3 Hz, 1H), 7.73–7.60 (m, 3H), 7.25 (d, J = 7.4 Hz, 1H), 7.20 (d, J = 2.3 Hz, 1H), 7.04–6.93 (m, 2H), 6.79–6.65 (m, 2H), 4.79–4.66 (m, 1H), 3.87–3.71 (m, 1H), 3.32– 3.26 (m, 1H), 3.23–3.02 (m, 5H), 2.91–2.78 (m, 2H), 1.49–1.11 (m, 24H); 13C NMR (100 MHz, DMSO-*d*_6_): δ 172.8, 171.6, 167.8, 165.1, 155.9, 138.9, 136.5, 135.3, 134.9, 132.6, 132.1, 131.6, 129.6, 129.1, 128.6, 128.4, 128.2, 127.7, 127.5, 124.0, 123.7, 123.1, 122.8, 121.4, 118.9, 118.7, 115.1, 111.8, 111.0, 78.4, 77.8, 55.1, 54.9, 39.0, 38.7, 33.5, 32.1, 29.7, 28.7, 28.6, 27.8, 23.3.; HRMS (ESI): *m*/*z* calcd for C_49_H_58_BrN_7_O_8_ [M + Na]+: 974.3422; found: 974.3420.

di-tert-butyl ((S)-6-((2-((S)-2-(2-([1,1′-biphenyl]-3-carboxamido)-5-bromobenzamido)-3-(1H-indol-3-yl)propanamido)ethyl)amino)-6-oxohexane-1,5-diyl)dicarbamate (**18b**).

The title compound **18b** was prepared from Boc-Lys(Boc)-Osu (0.142 g, 0.32 mmol) and **12j** (0.2 g, 0.32 mmol) according to the general procedure J. The product **18b** was obtained as an off-white solid (0.15 g, 56%); 1H NMR (400 MHz, DMSO-*d*_6_) δ 12.13 (s, 1H), 10.79 (s, 1H), 9.12 (d, J = 8.1 Hz, 1H), 8.52 (d, J = 8.9 Hz, 1H), 8.24 (s, 1H), 8.11 (s, 1H), 8.02 (s, 1H), 7.88 (q, J = 5.8, 7.5 Hz, 2H), 7.79–7.65 (m, 5H), 7.61 (t, J = 7.7 Hz, 1H), 7.50 (t, J = 7.6 Hz, 2H), 7.42 (dd, J = 6.2, 8.5 Hz, 1H), 7.26 (d, J = 7.9 Hz, 1H), 7.19 (s, 1H), 7.01 (t, J = 7.5 Hz, 1H), 6.94 (t, J = 7.5 Hz, 1H), 6.75–6.68 (m, 2H), 4.74–4.64 (m, 1H), 3.85–3.71 (m, 1H), 3.30–2.97 (m, 6H), 2.85 (d, J = 6.5 Hz, 2H), 1.66–1.42 (m, 2H), 1.35 (s, 18H), 1.31–1.16 (m, 4H);13C NMR (100 MHz, DMSO-*d*_6_) 172.8, 171.5, 167.7,164.9, 156.0, 155.7, 141.2, 139.7, 138.7, 136.5, 135.5, 135.2, 131.6, 130.8, 130.0, 129.6, 128.4, 127.6, 126.2, 125.8, 124.0, 123.1, 122.7, 121.3, 118.9, 118.7, 115.1, 111.7, 110.9, 78.4, 77.7, 55.1, 54.8, 39.9, 38.6, 29.6, 28.7, 28.6, 27.7, 23.2; HRMS (ESI): *m*/*z* calcd for C_49_H_58_BrN_7_O_8_ [M + Na]+: 974.3422; found: 974.3420.

Di-tert-butyl ((S)-6-((2-((S)-2-(2-([1,1′-biphenyl]-4-carboxamido)-5-bromobenzamido)-3-(1H-indol-3-yl)propanamido)ethyl)amino)-6-oxohexane-1,5-diyl)dicarbamate (**18c**).

The title compound **18c** was prepared from Boc-Lys(Boc)-Osu (0.142 g, 0.32 mmol) and **12k** (0.2 g, 0.32 mmol) according to the general procedure J. The product **18c** was obtained as an off-white solid (0.15 g, 50%);

1H NMR (400 MHz, DMSO-*d*_6_): δ 12.12 (s, 1H), 10.80 (d, J = 1.8 Hz, 1H), 9.14 (d, J = 8.0 Hz, 1H), 8.57 (d, J = 9.0 Hz, 1H), 8.28 (bs, 1H), 8.04 (d, J = 2.4 Hz, 1H), 7.93–7.80 (m, 5H), 7.79–7.73 (m, 3H), 7.70 (d, J = 7.7 Hz, 1H), 7.55–7.49 (m, 2H), 7.47–7.41 (m, 1H), 7.26 (d, J = 7.6 Hz, 1H), 7.20 (d, J = 2.2 Hz, 1H), 7.06–6.94 (m, 2H), 6.79–6.65 (m, 2H), 4.79–4.67 (m, 1H), 3.88–3.75 (m, 1H), 3.36– 3.27 (m, 1H), 3.21–3.08 (m, 5H), 2.90–2.79 (m, 2H), 1.49–1.13 (m, 24H); 13C NMR (100 MHz, DMSO-*d*_6_): δ 172.8, 171.5, 167.8, 164.6, 155.9, 144.1, 139.4, 138.9, 136.5, 135.3, 133.4, 131.6, 129.5, 128.7, 128.1, 127.7, 127.6, 127.4, 124.0, 122.8, 122.6, 121.3, 118.9, 118.7, 115.0, 111.8, 111.0, 78.4, 77.8, 55.1, 54.8, 39.1, 38.7, 32.1, 29.6, 28.7, 28.6, 27.8.; HRMS (ESI): *m*/*z* calcd for C_49_H_58_BrN_7_O_8_ [M + Na]+: 974.3422; found: 974.3418.

N-(4-bromo-2-(((S)-1-((2-((S)-2,6-diaminohexanamido)ethyl)amino)-3-(1H-indol-3-yl)-1-oxopropan-2-yl)carbamoyl)phenyl)-2-naphthamide (**19a**).

The title compound **19a** was prepared from compound **18a** (0.1 g, 0.10 mmol) according to the general procedure H. The product **19a** was obtained as an off-white solid (0.038 g, 49%);

1H NMR (600 MHz, DMSO-*d*_6_): δ 12.17 (d, J = 6.1 Hz, 1H), 10.84 (d, J = 2.0 Hz, 1H), 9.17 (dd, J = 8.0, 1.7 Hz, 1H), 8.58 (t, J = 4.5 Hz, 1H), 8.54 (dd, J = 8.9, 1.6 Hz, 1H), 8.44 (d, J = 1.0 Hz, 1H), 8.41–8.34 (m, 1H), 8.18 (bs, 3H), 8.08–8.00 (m, 4H), 7.85 (dd, J = 8.6, 1.8 Hz, 1H), 7.84–7.76 (m, 4H), 7.73–7.61 (m, 3H), 7.26 (d, J = 7.9 Hz, 1H), 7.22 (d, J = 1.7 Hz, 1H), 7.04–6.95 (m, 2H), 4.78–4.72 (m, 1H), 3.72–3.64 (m, 1H), 3.31 (td, J = 14.1, 3.2 Hz, 1H), 3.28–3.08 (m, 5H), 2.79–2.69 (m, 2H), 1.75–1.63 (m, 2H), 1.56–1.47 (m, 2H), 1.36–1.25 (m, 2H); 13C NMR (150 MHz, DMSO-*d*_6_): δ 171.7, 169.0, 167.8, 165.1, 158.7, 138.8, 136.5, 135.3, 134.9, 132.6, 132.0, 131.6, 129.6, 129.1, 128.7, 128.4, 128.2, 127.6, 127.6, 124.1, 123.7, 123.2, 123.1, 122.9, 121.4, 118.9, 118.7, 118.4, 116.4, 115.2, 111.8, 110.9, 55.1, 52.6, 38.9, 38.8, 38.6, 30.8, 27.8, 27.0, 21.7, 21.7.; HRMS (ESI): *m*/*z* calcd for C_37_H_40_BrN_7_O_4_ [M + H]+: 726.2396; found: 726.2395.

N-(4-bromo-2-(((S)-1-((2-((S)-2,6-diaminohexanamido)ethyl)amino)-3-(1H-indol-3-yl)-1-oxopropan-2-yl)carbamoyl)phenyl)-[1,1′-biphenyl]-3-carboxamide (**19b**)

The title compound **19ab** was prepared from compound **18b** (0.1 g, 0.10 mmol) according to the general procedure H. The product **19b** was obtained as an off-white solid (0.041 g, 55%);

1H NMR (600 MHz, DMSO-*d*_6_): δ 12.11 (s, 1H), 10.83 (d, J = 1.7 Hz, 1H), 9.14 (d, J = 7.9 Hz, 1H), 8.56 (bs, 1H), 8.49 (d, J = 8.9 Hz, 1H), 8.35–8.31 (m, 1H), 8.20–7.80 (bs, 5H), 8.12 (t, J = 8.9 Hz, 1H), 8.01 (d, J = 2.3 Hz, 1H), 7.94–7.90 (m, 1H), 7.79–7.70 (m, 5H), 7.68 (d, J = 7.9 Hz, 1H), 7.54–7.50 (m, 2H), 7.46–7.41 (m, 1H), 7.28 (d, J = 8.1 Hz, 1H), 7.21 (d, J = 2.2 Hz, 1H), 7.05–7.00 (m, 1H), 6.98–6.93 (m, 1H), 4.76–4.67 (m, 1H), 3.67 (t, J = 6.4 Hz, 1H), 3.39– 3.28 (m, 1H), 3.21–3.10 (m, 5H), 2.75 (t, J = 7.8 Hz, 2H), 1.75–1.63 (m, 2H), 1.57–1.48 (m, 2H), 1.35–1.27 (m, 2H); 13C NMR (150 MHz, DMSO-*d*_6_): δ 171.7, 169.1, 167.8, 165.0, 158.5, 141.3, 139.7, 138.7, 136.5, 135.5, 135.3, 131.6, 130.8, 130.1, 129.6, 128.5, 127.6, 127.3, 126.3, 125.9, 124.0, 123.3, 122.9, 121.4, 118.8, 118.7, 116.7, 115.2, 111.8, 110.9, 55.1, 52.6, 38.9, 38.7, 38.6, 30.9, 27.8, 27.0, 21.7.; HRMS (ESI): m/z calcd for C_39_H_42_BrN_7_O_4_ [M + H]+: 752.2554; found: 752.2553.

N-(4-bromo-2-(((S)-1-((2-((S)-2,6-diaminohexanamido)ethyl)amino)-3-(1H-indol-3-yl)-1-oxopropan-2-yl)carbamoyl)phenyl)-[1,1′-biphenyl]-4-carboxamide (**19c**).

The title compound **19c** was prepared from compound **18c** (0.1 g, 0.10 mmol) according to the general procedure H. The product **19c** was obtained as an off-white solid (0.031 g, 49%); 1H NMR (600 MHz, DMSO-*d*_6_): δ 12.08 (s, 1H), 10.83 (d, J = 2.1 Hz, 1H), 9.16 (d, J = 8.1 Hz, 1H), 8.58 (t, J = 5.2 Hz, 1H), 8.55 (d, J = 9.0 Hz, 1H), 8.37 (t, J = 5.2 Hz, 1H), 8.16 (s, 3H), 8.04 (d, J = 2.4 Hz, 1H), 7.89–7.86 (m, 2H), 7.85–7.82 (m, 2H), 7.81–7.72 (m, 6H), 7.69 (d, J = 7.7 Hz, 1H), 7.55–7.51 (m, 2H), 7.47–7.43 (m, 1H), 7.27 (d, J = 8.0 Hz, 1H), 7.21 (d, J = 2.3 Hz, 1H), 7.05–7.01 (m, 1H), 7.00–6.96 (m, 1H), 6.58 (bs, 1H), 4.79–4.72 (m, 1H), 3.72–3.64 (m, 1H), 3.35–3.29 (m, 1H), 3.25–3.12 (m, 5H), 2.79–2.70 (m, 2H), 1.75–1.64 (m, 2H), 1.56–1.47 (m, 2H), 1.36–1.26 (m, 2H); 13C NMR (150 MHz, DMSO-*d*_6_): δ 171.7, 169.0, 167.8, 164.6, 158.6, 144.1, 139.3, 138.9, 136.5, 135.4, 133.4, 131.6, 129.6, 128.8, 128.1, 127.7, 127.6, 127.4, 124.1, 122.8, 122.7, 121.4, 118.9, 118.7, 115.0, 111.9, 110.9, 55.1, 52.6, 38.9, 38.8, 38.6, 30.8, 27.8, 27.0, 21.7.; HRMS (ESI): *m*/*z* calcd for C_39_H_42_BrN_7_O_4_ [M + H]+: 752.2554; found: 752.2551.

## 5. Conclusions

In summary, three series of anthranilamide peptidomimetic compounds were developed by varying the nature and position of the hydrophobic group and the cationic charge of the molecules. These led to the synthesis of novel compounds showing potent antibacterial activity against *S. aureus* and moderate activity against *E. coli*. These compounds also varied in their toxicity to mammalian cells, with the series III compounds showing the best overall selectivity for both bacterial strains over human cells. The compound 19c showed more than 50% of *S. aureus* biofilm disruption at 31.2 µM and non-toxic to mammalian cells at this concentration. Results from the membrane permeability assay and membrane integrity experiment with *B. subtilis* demonstrated that the active compounds could act via cell membrane disruption. Importantly, these compounds are also found to eradicate established bacterial biofilms of *S. aureus* and *E. coli*. Hence, this class of peptidomimetic compounds represents an innovative avenue for the development of effective antibacterial and antibiofilm agents.

## Data Availability

Data are contained within the article and Appendix A.

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
