# Peer review of "Tuning the Anthranilamide Peptidomimetic Design to Selectively Target Planktonic Bacteria and Biofilm"

_antibiotics, 2023, doi:10.3390/antibiotics12030585_

Round 1
Reviewer 1 Report
Please, review the format of some references.
Author Response
Dear Reviewer,
Thank you for your valuable comment. I have formatted all the references according to the MDPI.
Reviewer 2 Report
This manuscript by Kuppusamy et al. reported the discovery of short amphiphilic anti-bacterial and antibiofilm agents produced by tuning the hydrophobic and cationic groups of anthranilamide peptidomimetics. Among the designed and synthesized compounds, 12j and 16b displayed the best anti-bacterial activities with IC50 values of 3.9 μM. Compound 19c showed more than 50% of S. aureus biofilm disruption at 31.2 μM. Overall, the manuscript is well organized and written, and may be useful to the related research field. This reviewer recommends the acceptance for publication in Antibiotics after the following concerns are addressed.
1. The author highlighted the series III compounds, but their inhibitory activity against S. aureus and E. coli are moderate (15.6 μM≈12 μg/mL?). The strategy behind (introduction of a lysine cationic group) is known, and the results are ordinary. For table 1, no positive control was provided which compromised the significance of the work. Are the compounds superior than the positive control (e.g., colistin) or not? If not, why they are useful for the relevant research field?
2. Is it necessary to list Table 2, given the contents are exactly the same with Table 1.
3. Figure 4, why compound 19a with a same lysine cationic group is less able to penetrate membrane than 19b?
4. Line 196, compound 19c should be 19b.
5. Figure 5, B. subtilis was used instead of S. aureus or E. Coli, why?
6. Figure 6 and Figure 7, significant difference should be added.
7. More recent references should be cited.
Author Response
Dear Reviewer,
Thank you for your valuable comments.
We have addressed everything as shown below.
This manuscript by Kuppusamy et al. reported the discovery of short amphiphilic anti-bacterial and antibiofilm agents produced by tuning the hydrophobic and cationic groups of anthranilamide peptidomimetics. Among the designed and synthesized compounds, 12j and 16b displayed the best anti-bacterial activities with IC50 values of 3.9 μM. Compound 19c showed more than 50% of S. aureus biofilm disruption at 31.2 μM. Overall, the manuscript is well organized and written, and may be useful to the related research field. This reviewer recommends the acceptance for publication in Antibiotics after the following concerns are addressed.
- The author highlighted the series III compounds, but their inhibitory activity against S. aureus and E. coli are moderate (15.6 μM≈12 μg/mL?). The strategy behind (introduction of a lysine cationic group) is known, and the results are ordinary. For table 1, no positive control was provided which compromised the significance of the work. Are the compounds superior than the positive control (e.g., colistin) or not? If not, why they are useful for the relevant research field?
The manuscript included several compounds that had activity between 3.9 to 7.8 μM against S. aureus. The strategy of introducing a lysine was to reduce the cytotoxicity; and the data showed that thins reduced toxicity as the compound was not toxic even upto 160 μM.
We have tested colistin, and found that our peptidomimetic compounds have a lower MIC with S. aureus 38 than colistin (NOW ADDED TO TABLE 1). Colistin’s activity was greater than our compounds against E. coli (TABLE 1). However colistin is known to induced nephrotoxicity and we have added this into the discussion (SEE PAGE 8, LINE182-185).
- Is it necessary to list Table 2, given the contents are exactly the same with Table 1.
We prefer to keep this in the papers as it helps to understand the difference in MIC against S. aurus and E.coli after addition of lysine.
- Figure 4, why compound 19a with the same lysine cationic group is less able to penetrate membrane than 19b?
Though the cationic group is similar the hydrophobic group is different ( Naphthyl Vs Biphenyl) and it may play a role in less membrane permeability. We have added this into the discussion at SEE PAGE 8, LINE199-201.
- Line 196, compound 19c should be 19b.
The typo error was fixed and changed to 19b.
- Figure 5, subtiliswas used instead of S. aureus or E. Coli, why?
We were fortunate to have a strain of B. subtilis BS23 that had GFP labelled the α-subunit of the membrane-localized ATP synthase that would allow us to demonstrate the direct effect of our compounds on the bacterial membrane. There is no such labelled strain for S. aureus and E.coli inhouse.
- Figure 6 and Figure 7, significant difference should be added.
We have changed the MIC values into actual numbers to make the figures more meaningful. Thank you for the comment.
- More recent references should be cited.
Recent references were cited.
- Teng, P.; Shao, H.; Huang, B.; Xie, J.; Cui, S.; Wang, K.; Cai, J. Small Molecular Mimetics of Antimicrobial Peptides as a Promising Therapy To Combat Bacterial Resistance. Journal of Medicinal Chemistry 2023, 66, 2211-2234, doi:10.1021/acs.jmedchem.2c00757.
- Haidari, H.; Melguizo-Rodríguez, L.; Cowin, A.J.; Kopecki, Z. Therapeutic potential of antimicrobial peptides for treatment of wound infection. American Journal of Physiology-Cell Physiology 2023, 324, C29-C38, doi:10.1152/ajpcell.00080.2022.
and also some new references were updated.
Reviewer 3 Report
Reviewer comments
Manuscript ID: antibiotics-2264288
Title: Tuning the anthranilamide peptidomimetic design to selectively target planktonic bacteria and biofilm
The present manuscript authored by Rajesh et.al., describes the synthesis of novel anthranilamide derivatives and their biological activities. This work aims to target both planktonic bacteria and biofilm in a selective manner. The work was planned executed and described in a standard manner and work seems to be reliable and reproducible. Based on these points, the present manuscript can be acceptable with the following minor revisions.
1. Author has to elaborate the necessity to use hazardous solvents such as DCM, DMF and THF. What about non hazardous solvents to carry out the reactions.
2. A series of synthetic steps were optimized in these synthetic methodologies. Is there any one pot reaction or multi-component reactions kind of methodologies was studied?
3. To describe the importance of biologically active amides, the following reference may be included in the reference section.
https://doi.org/10.1016/j.tetasy.2011.03.009
Author Response
Dear Reviewer,
Thank you for your valuable comment. We have addressed the comments as shown below.
Title: Tuning the anthranilamide peptidomimetic design to selectively target planktonic bacteria and biofilm
The present manuscript authored by Rajesh et.al., describes the synthesis of novel anthranilamide derivatives and their biological activities. This work aims to target both planktonic bacteria and biofilm in a selective manner. The work was planned executed and described in a standard manner and work seems to be reliable and reproducible. Based on these points, the present manuscript can be acceptable with the following minor revisions.
- Author has to elaborate the necessity to use hazardous solvents such as DCM, DMF and THF. What about nonhazardous solvents to carry out the reactions.
Hazardous solvents such as dichloromethane (DCM), dimethylformamide (DMF), and tetrahydrofuran (THF) are often used in organic reactions due to their ability to dissolve a wide range of reactants and provide a suitable reaction environment. For example, in amide bond formation, the effective dissolution of the coupling agent is attained to enhance the reaction. The reaction kinetics of acid-amine coupling reactions is highly dependent on the reaction environment, including the solvent used. For example, DCM is known to increase the reaction rate due to its high polarity and low boiling point, which can help drive off water and promote the formation of the amide bond. Similarly, DMF and THF can also enhance the reaction rate and provide a suitable reaction environment. In the formation of acid chloride, DMF is important reagent to activate the oxalyl chloride. For the Suzuki–Miyaura coupling, methyl t-butyl ether (MTBE), cyclopentyl methyl ether (CPME), diethyl carbonate (DEC), p-cymene, dimethylcarbonate (DMC), and anisole as alternative recommended solvents in terms of health, safety, and environmental ranking for the coupling. However, for these solvents, the solubility is not good for the reactants used in this paper. Also, the simple, widely used palladium catalysts such as tetrakis is not suitable to the listed alternative solvent. Safer alternatives such as ionic liquids or water-based solvents should be considered however the solubility of the reactants is very poor.
- A series of synthetic steps were optimized in these synthetic methodologies. Is there any one pot reaction or multi-component reactions kind of methodologies was studied?
We did not perform any one-pot or multi-component reactions in this synthesis route.
- To describe the importance of biologically active amides, the following reference may be included in the reference section.
We have added a few more new and recent references to the manuscript. We also added the reference https://doi.org/10.1016/j.tetasy.2011.03.009 mentioned by the reviewer.
Round 2
Reviewer 2 Report
Most concerns have been explained or addressed. Can be accepted.